# Integrating Planning and Deep Reinforcement Learning via Automatic Induction of Task Substructures

**Jung-Chun Liu, Chi-Hsien Chang, Shao-Hua Sun, Tian-Li Yu**
Department of Electrical Engineering
National Taiwan University
{r10921043,d07921004,shaohuas,tianliyu}@ntu.edu.tw

## Abstract

Despite recent advancements, deep reinforcement learning (DRL) still struggles at learning sparse-reward goal-directed tasks. Classical planning excels at addressing hierarchical tasks by employing symbolic knowledge, yet most of the methods rely on assumptions about pre-defined subtasks. To bridge the best of both worlds, we propose a framework that integrates DRL with classical planning by automatically inducing task structures and substructures from a few demonstrations. Specifically, genetic programming is used for substructure induction where the program model reflects prior domain knowledge of effect rules. We compare the proposed framework to state-of-the-art DRL algorithms, imitation learning methods, and an exploration approach in various domains. Experimental results show that our proposed framework outperforms all the abovementioned algorithms in terms of sample efficiency and task performance. Moreover, our framework achieves strong generalization performance by effectively inducing new rules and composing task structures. Ablation studies justify the design of our induction module and the proposed genetic programming procedure.

## 1 Introduction

Deep reinforcement learning (DRL) as an inductive learning method allows agents to deal with high-dimensional decision-making problems considered intractable in the past (Arulkumaran *et al.*, 2017). DRL has applied to various fields, including robotics (Nguyen *et al.*, 2020), autonomous driving (Kiran *et al.*, 2022), and video games (Mnih *et al.*, 2013). However, exploring complex tasks with sparse and delayed rewards still remains challenging, leading to inapplicability on many real-world problems comprising multiple subtasks, *e.g.*, cooking and furniture assembly.

In contrast, classical planning is a deductive learning method which aims to solve planning and scheduling problems. Particularly, classical planning is adept at finding the sequence of actions in deterministic and known environments. Researchers in classical planning have developed effective planners that can handle large-scale problems (Vallati *et al.*, 2015). Yet, classical planning agents face difficulties exploring environments due to limitations in model and domain-specific representation in unknown environments where action models are undiscovered.

Several methods work on combining planning and DRL to address hierarchical tasks with high-level abstraction. Konidaris *et al.* (2018) develop a skill-up approach to build a planning representation from skill to abstraction, while they do not encompass skill acquisition from low-level execution. Mao *et al.* (2022) introduce an extension of planning domain definition language (Ghallab *et al.*, 1998) to model the skill, and Silver *et al.* (2022) propose a method for learning parameterized policies integrated with symbolic operators and neural samplers. However, they consider object-centric representations, which require fully observable environments and carefully designed predicates.

In this paper, we combine classical planning and DRL to augment agents effectively to adapt to environments by inducing underlying prior knowledge from expert demonstrations. Specifically, we devise a method that induces symbolic knowledge using genetic programming (Koza, 1994), an evolutionary computation approach, to discover task substructures that accurately capture the underlying patterns within the data. The compositional property of the programs enables generalizability that adapts to new environments by discovering new substructures from known ones.

To evaluate the proposed framework, we design three gridworld environments where agents can move on and interact with objects. The result shows the improvement of DRL agents and outperformance compared to other imitation learning and exploration-based methods. Also, our framework demonstrates generalizability by inducing variant substructures and recomposing task structures. Finally, we show the ablation studies about the accuracy of induction.

## 2 RELATED WORK

**Learning abstraction from demonstrations.** State abstraction facilitates the agent's reasoning capabilities in high-level planning by extracting symbolic representations from low-level states (Abel *et al.*, 2018; Guan *et al.*, 2022). Agents can learn the abstraction from demonstrations since demonstrations encompass valuable information regarding task composition and relevant features (Byrne & Russon, 1998). Some methods were developed to extract task decomposition and abstraction from demonstrations (Hayes & Scassellati, 2016; Cobo *et al.*, 2011). Our work extends these approaches to infer knowledge from demonstrations.

**Learning planning action models.** Some works have focused on integrating learning and planning to enhance capabilities in complex environments (Danesh *et al.*, 2023; Veloso *et al.*, 1995). To leverage the strategies of classical planning, many works have developed building planning action models, including skill acquisition and action schema learning (Arora *et al.*, 2018; Stern & Juba, 2017; Pasula *et al.*, 2007; Callanan *et al.*, 2022; Silver *et al.*, 2022; Mao *et al.*, 2022; Yang *et al.*, 2007). However, these works mainly focus on the existing planning benchmark and less focus on general Markov decision process (MDP) problems. On the other hand, to address the issue of sample efficiency in DRL, several techniques explored the integration of symbolic knowledge with DRL by learning planning action models (Jin *et al.*, 2022; Lyu *et al.*, 2019). In this work, we aim to bridge this gap by extending these approaches to incorporate inferred knowledge into DRL, enhancing its applicability in complex decision-making scenarios.

**Hierarchical task learning.** A proper hierarchical structure is crucial for task decomposition and abstraction. Various methods have been proposed for constructing hierarchical task representation (Pateria *et al.*, 2022), including graphs (Svetlik *et al.*, 2017), automata (Furelos-Blanco *et al.*, 2021; Icarte *et al.*, 2019; 2022; Xu *et al.*, 2021), programs (Sun *et al.*, 2018; 2020; Trivedi *et al.*, 2021; Liu *et al.*, 2023; Lin *et al.*, 2023), and hierarchical task networks (Hayes & Scassellati, 2016; Sohn *et al.*, 2020; Lee *et al.*, 2019). Some approaches utilize the capabilities of deep learning with intrinsic rewards (Kulkarni *et al.*, 2016; Florensa *et al.*, 2017). In addition, some of these works specifically address leveraging knowledge to deal with multiple compositional tasks via task decomposition (Andreas *et al.*, 2017; Sohn *et al.*, 2022; Liu *et al.*, 2022; Sun *et al.*, 2020; Sohn *et al.*, 2018; Furelos-Blanco *et al.*, 2021). Despite the success in building hierarchical models shown in previous works, these works put less emphasis on inducing subtask rules and substructure. Therefore, we develop a method to induce symbolic knowledge and leverage it for hierarchical task representation.

## 3 PROBLEM FORMULATION

We address the sparse-reward goal-directed problems which can be formulated as MDPs denoted as $\langle \mathcal{S}, \mathcal{A}, T, R, \gamma \rangle$, where $\mathcal{S}$ denotes state space, $\mathcal{A}$ denotes action space, $T : \mathcal{S} \times \mathcal{A} \to \mathcal{S}$ denotes a transition function, $R : \mathcal{S} \times \mathcal{A} \to \mathbb{R}$ denotes a reward function, and $\gamma \in (0, 1]$ denotes a discounting factor. DRL agents often struggle at solving sparse-reward, hierarchical tasks, while classical planning techniques excel in such scenarios. On the other hand, unlike classical planning, DRL, as a generic model-free framework, does not require pre-defined models. This motivates us to bridge the best of both worlds by integrating these two paradigms.

However, while DRL directly learns from interacting with MDPs, classical planning operates on literal conjunctions. To address this gap, we integrate planning and DRL methods by annotating the specific MDP actions in the form of action schemata. Specifically, we consider the problem of inducing the action schemata from demonstrations. The objective is to induce the action schemata which can be leveraged for task structure deduction. After the action schemata are discovered, the framework deduces task structures from the action model and aims to offer guidance for the training of DRL agents based on the task structures.

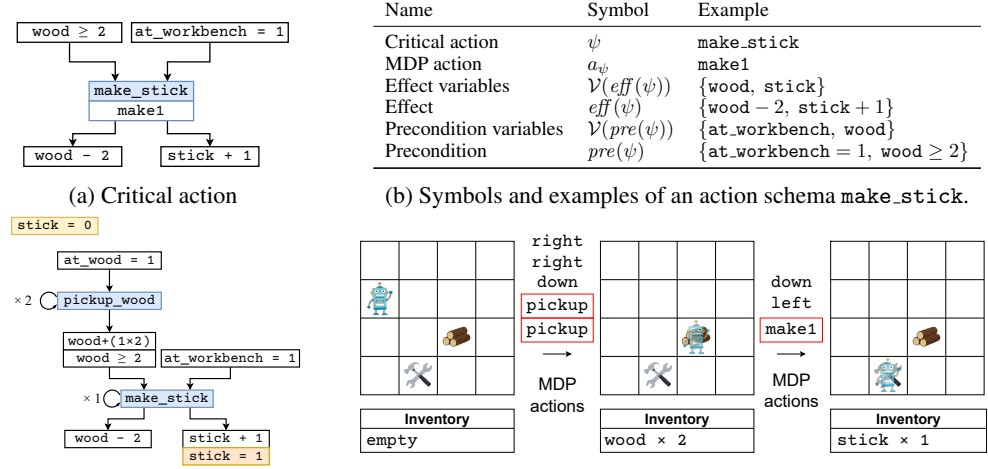

(a) Critical action

(b) Symbols and examples of an action schema make_stick.

| Name | Symbol | Example |
|---|---|---|
| Critical action | $\psi$ | make_stick |
| MDP action | $a_\psi$ | make1 |
| Effect variables | $\mathcal{V}(\textit{eff}(\psi))$ | {wood, stick} |
| Effect | $\textit{eff}(\psi)$ | {wood $- 2$, stick $+ 1$} |
| Precondition variables | $\mathcal{V}(\textit{pre}(\psi))$ | {at_workbench, wood} |
| Precondition | $\textit{pre}(\psi)$ | {at_workbench $= 1$, wood $\geq 2$} |

(c) Critical action network

(d) Example of making a stick in MINECRAFT

Figure 1: **Critical action. (a)-(b) The illustration, symbols, and examples of a critical action.** A critical action is an essential action in environments with preconditions and effects. **(c) Critical action network.** If an action model is discovered by the induction module, it builds critical action networks. **(d) Example of making a stick in MINECRAFT.** Actions highlighted by red rectangles are critical, *i.e.*, picking up wood twice and making a stick (make1).

## 4 INTEGRATION OF MDP IN DRL AND PLANNING WITH CRITICAL ACTION

To bridge the gap between DRL and planning, we introduce the concept of critical actions in this section. Specifically, we formulate our problems as mapping tasks described by MDPs to planning domain definition language (PDDL) (Ghallab *et al.*, 1998; Silver & Chitnis, 2020) and SAS⁺ (Bäckström & Nebel, 1995), where the preliminary notation is elaborated in Appendix A. To express numeric variables, we adopt the configuration of PDDL 2.1 (Fox & Long, 2003), which includes arithmetic operators for specification.

**Mapping between MDP and classical planning.** Lee *et al.* (2021) has developed the abstraction mapping between planning and MDP problems. A planning task $\Pi = \langle \mathcal{V}, \mathcal{O}, s'_g \rangle$, where $\mathcal{V}$ is a set of variables, $\mathcal{O}$ is a set of operators in the domain, and $s'_g$ is a partial state describing the goal. The transition graph of the planning task is a tuple $\mathcal{T}' = \langle \mathcal{S}', T', \mathcal{S}'_{goal} \rangle$, where $\mathcal{T}'$ is a set of transitions $\langle s', o, T'(s') \rangle$ for all $s'$ in $\mathcal{S}'$, and $\mathcal{S}'_{goal}$ is a set of goal states. Let $\mathcal{L} : \mathcal{S} \to \mathcal{S}'$ be a mapping from the MDP state space $\mathcal{S}$ to high-level planning state space $\mathcal{S}'$. Given an MDP problem, the abstraction $\langle \mathcal{L}, \Pi \rangle$ is proper iff there exists a mapping $\mathcal{L}$ to $\Pi$ such that $\langle \mathcal{L}(s), \psi, \mathcal{L}(T(s, a_\psi)) \rangle \in \mathcal{T}'$ if some $\psi$ is admissible in the MDP state $s \in \mathcal{S}$ or $\mathcal{L}(s) = \mathcal{L}(T(s, a_\psi))$, where $\mathcal{T}'$ is a set of all possible transitions in $\Pi$. In this work, we focus on the MDP problems with proper abstraction in which a mapping to a planning domain exists, and action models can be induced by the proposed framework.

**Critical action.** Critical actions are the actions that lead to the transitions in $\mathcal{T}'$. That is, these actions are critical for progress at the planning level and must be executed in a specific order. A state $s' \in \mathcal{S}'$ in the planning domain is an assignment to $\mathcal{V}$, and $s'_v \in \mathbb{R}$ is the value assigned to the variable $v \in \mathcal{V}$. We map each $s_i$ in MDP problems with distinct $s'_v$ in planning, considering a planning task as an MDP-like tuple $\langle \mathcal{S}', \mathcal{O}, T' \rangle$, a transition $\langle s, a_\psi, T(s, a_\psi) \rangle$ in an MDP problem can be directly transferred to the transition $\langle \mathcal{L}(s), \psi, \mathcal{L}(T(s, a_\psi)) \rangle$ in planning domain.

- **MDP action** $a_\psi \in \mathcal{A}$ denotes the MDP action mapping to $\psi$.
- **Precondition** $\textit{pre}(\psi)$ is a set of conditions requires satisfaction before executing $\psi$.
- **Effect** $\textit{eff}(\psi)$ is a set of functions which indicates the state change after executing $\psi$.

A state $s' \in \mathcal{S}'$, where $\mathcal{S}'$ denotes planning state space, is an assignment to $\mathcal{V}$, where $\mathcal{V}(p)$ denotes the variables of the assignment $p$. Given an effect $\textit{eff}(\psi)$ and one of its variables $v \in \mathcal{V}(\textit{eff}(\psi))$,

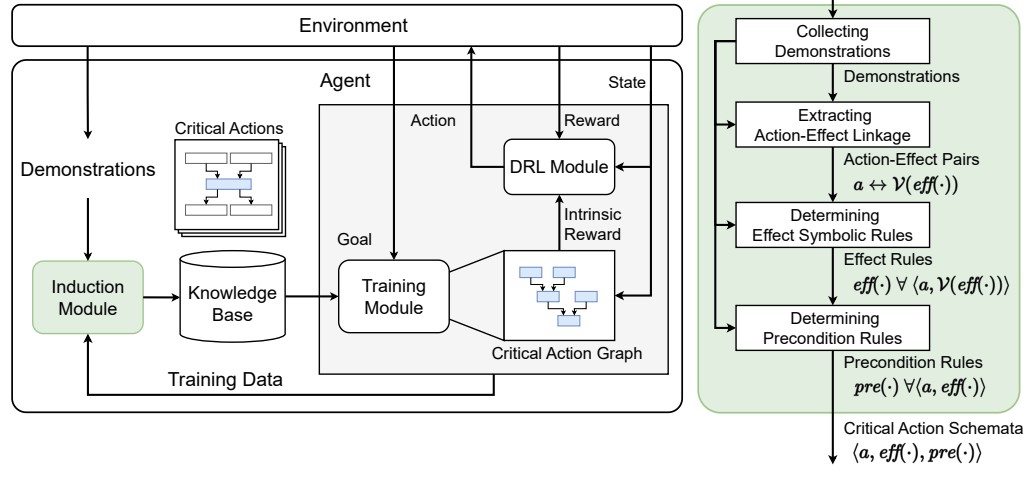

(a) Framework overview          (b) Induction module

Figure 2: **(a) Framework overview.** The proposed framework is two-stage. In the *induction stage*, critical action schemata are induced from demonstrations. In the *training stage*, the training module deduces the critical action network from the goal by backward-chaining and offers intrinsic rewards to the DRL module according to the network. **(b) Induction module.** The induction module induces the critical action schemata from demonstrations through three steps. First, it finds the linkage between actions and effect variables in transitions. Then, given the transitions with action-effect linkage, the induction module induces the effect rules via symbolic regression. Finally, it determines the precondition given the specific action and the effect.

an effect rule $eff(\psi)_v : \mathbb{R} \to \mathbb{R}$ is a function which transfers the specific feature value $s'_v$ in $s'$ to another value $eff(\psi)_v[s'_v]$ in the transition, and a precondition rule $pre(\psi)_v$ is a logical formula $pre(\psi)_v : \mathbb{R} \to \{0, 1\}$ that determines whether the variable $v$ is satisfied to execute $\psi$. Given a state $s'$, two critical action $\psi$ and $\phi$, $eff(\psi)_v$ satisfy $pre(\phi)_v$ in state $s'$ iff a variable $v$ in both $\mathcal{V}(eff(\psi))$ and $\mathcal{V}(pre(\phi))$, and $pre(\phi)_v[s'_v]$ is false while $pre(\phi)_v[eff(\psi)_v[s'_v]]$ is true in a transition with $\psi$. That is, executing $\psi$ makes $\phi$ become admissible.

To efficiently induce the model, we assume that the properties of the features in a state are known. Effect variable space $\mathbb{E} = \{v \mid v \in \mathcal{V}(eff(\psi)) \, \forall \, \psi \in \mathcal{O}\}$ contains the variables that will change in transitions and related to the progress of the tasks.

**Critical action network.** This work represents symbolic knowledge structures as critical action networks illustrated in Figure 1c. Given a set of critical actions $\mathcal{O}$ and a desired goal specification $p_{goal}$, a critical action network $\mathcal{G} = (V, E)$ is an in-tree structure where the root is the critical action that can satisfy the goal specification directly. For each edge $(\psi, \phi) \in E$, there exists $eff(\psi)_v$ for some $v$ that satisfy $pre(\phi)_v$. Once the action schemata are known, we can construct the network using planners or backward chaining.

## 5   METHOD

We introduce the induction module that determines the critical actions from demonstrations and extracts symbolic rules in Section 5.1. Then, Section 5.2 describes the training module that deduces task structures to build critical action networks online from the given goal. The network contains subtask dependencies, providing guidance through intrinsic rewards and augmenting the training efficiency of DRL agents. An overview of our proposed framework is illustrated in Figure 2a.

### 5.1   INDUCTION MODULE

The procedure of the induction module is illustrated in Figure 2b. The module first extracts action-effect linkages $(a, \mathcal{V}(eff(\psi)))$ from demonstrations. Second, the module induces effect rules $eff(\psi)$ given $(a, \mathcal{V}(eff(\psi)))$. Finally, the module leverages the rules to determine the precondition rules

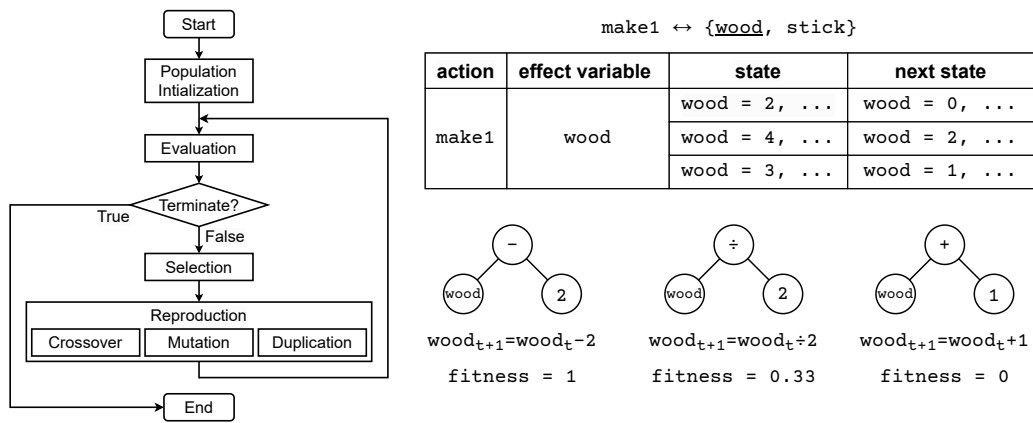

(a) Procedure of genetic programming          (b) Example of fitness evaluation

Figure 3: **Symbolic regression using genetic programming.** Given a pair of an MDP action and effect variables, symbolic regression is used to determine the rules when executing the action. **(a) Procedure of genetic programming.** The programs iteratively evolve through fitness evaluation, selection, and reproduction. **(b) Example of fitness evaluation.** The algorithm evaluates the accuracy of programs to induce the rule between `make1` and `wood`.

$pre(\psi)$ for each $(a, \mathit{eff}(\psi))$. After these steps, the components of critical action schemata are all determined. Note that we name the critical action $\psi$ for the convenience of reference, which is not known when inducing action schemata. We use "·" to represent an undefined critical action. The following paragraphs will elaborate on the details of the induction methods.

**Action-effect linkages.** Based on the outcome assumption that one action only impacts specific state features, we can detect co-occurrence of what effects often occur after executing $a$ by calculating mutual information (Shannon, 1948) between actions and effect variables. Let $\mathcal{E}$ be a set of possible effect variable combinations $\mathcal{V}(\mathit{eff}(\cdot))$ in the transitions of demonstrations. The mutual information $M_{(a, \mathcal{V}(\mathit{eff}(\cdot)))}$ is defined as follows:

$$M_{(a, \mathcal{V}(\mathit{eff}(\cdot)))} = \sum_{a \in \mathcal{A}} \sum_{\mathcal{V}(\mathit{eff}(\cdot)) \in \mathcal{E}} P_{\mathcal{A}\mathcal{E}}(a, \mathcal{V}(\mathit{eff}(\cdot))) \log \frac{P_{\mathcal{A}\mathcal{E}}(a, \mathcal{V}(\mathit{eff}(\cdot)))}{P_{\mathcal{A}}(a) P_{\mathcal{E}}(\mathcal{V}(\mathit{eff}(\cdot)))}, \tag{1}$$

where $P_{\mathcal{A}}(a)$ is the count of transitions with action $a$; $P_{\mathcal{E}}(\mathcal{V}(\mathit{eff}(\cdot)))$ is the count of transitions that include variables in $\mathcal{V}(\mathit{eff}(\cdot))$; $P_{\mathcal{A}\mathcal{E}}(a, \mathcal{V}(\mathit{eff}(\cdot)))$ is the count of transitions that include changed variables in $\mathcal{V}(\mathit{eff}(\cdot))$ with action $a$. To determine the linkage, the pairs are divided into two clusters with the threshold of a maximum gap, and the cluster with higher values are selected. The detailed algorithm is shown in Appendix B.1.

**Effect symbolic rules.** Given an action-effect pair $(a, \mathcal{V}(\mathit{eff}(\cdot)))$, the induction module proceeds to search for the effect $\mathit{eff}(\cdot)$, which can be formulated as a symbolic regression. To accomplish this, we employ genetic programming for symbolic regression to discover each effect rule $\mathit{eff}(\cdot)_v$ for all $v$ in $\mathcal{V}(\mathit{eff}(\cdot))$, aiming to discover programs that can accurately predict the effects.

In genetic programming, each program is represented as an expression tree, taking $s_v$ and $a_\psi$ in each transition as input and yielding the predicted value of $v$ after the transition as output. The algorithm consists of three key steps: initialization, evaluation, selection, and reproduction. Initially, a population of programs is randomly generated. The fitness of each program is evaluated based on its prediction accuracy, and the programs with the highest fitness values are selected, serving as parents to reproduce offspring through crossover, mutation, and duplication mechanisms. The procedures and the example of genetic programming are illustrated in Figure 3.

The model of symbolic rules is regarded as the substructures of the subtasks, and selecting the proper operators for the symbolic model compatible with the effects plays a crucial role in facilitating effective inference. For instance, in the context of general DRL task with numerical variable representation configuration, arithmetic operation set $\mathcal{F} = \{+, -, \times, \div, \mathrm{inc}, \mathrm{dec}\}$ is used as the function set in genetic programming, where $\mathrm{inc}$ denotes an increment operator and $\mathrm{dec}$ denotes a decrement oper-

ator. This choice of function set is consistent with the numerical variable representation commonly employed in DRL tasks. The underlying assumption guiding our approach is that the effects can be expressed through these programs, serving as prior knowledge of the problem. This allows our method to induce task substructures and generalize the knowledge across domains that share identical operation configurations. This distinguishing feature sets our approach apart from alternative model-free methodologies. Additional implementation details can be found in Appendix B.2.

**Precondition rules.** After the relation between $a$ and $eff(\cdot)$ are found, determining precondition rules $pre(\cdot)$ can be formulated as a classification problem, as the objective is to identify whether $eff(\cdot)$ occurs given the action and the state. The process involves minimal consistent determination (MCD) and the decision tree method. The model of $pre(\cdot)$ decides what preconditions leading to desired effects after executing $a$. Additional details can be found in Appendix B.3.

## 5.2 TRAINING MODULE

After the induction process, the critical action schemata serve as the components of knowledge base that guides the agent in the training stage. During the training stage, the training module deduces the critical action network given the initial state and goal specification and provides intrinsic reward if the agent successfully performs an action that meets the critical effects in the network.

**Inferring critical action network.** Once the critical actions schemata are defined, we can infer task structures from the model. Given a goal and an initial state, the proposed framework deduces the critical action networks by backward chaining. Starting from the goal, the module searches for the critical action to find the desired effect for unconnected precondition rules $pre(\cdot)_v$ where $v \in \mathbb{E}$. Maximum operation steps are set to terminate the search. Once the critical action is found, the critical action will be considered as the predecessor of previous critical actions.

**DRL agent.** In the training stage, we aim to train a DRL agent that can learn the subtask by leveraging the feature-extracting power of neural networks. The induction module only specifies the coarse-grained critical action to express temporal order. Therefore, the framework deploys DRL to complete the fine-grained decision-making tasks, which utilizes deep learning to approximate the optimal policy with neuron networks. DRL uses the policy gradient method to update the policy. In the proposed method, we use the action-critic method (Mnih *et al.*, 2016; Raffin *et al.*, 2021) as the DRL agent. The implementation details are described in Appendix B.4.

**Intrinsic rewards.** During the training stage, if the agent successfully executes the critical effects, it will receive an intrinsic reward when the preconditions of a critical action $\psi$ are satisfied. Conversely, if the agent takes an action that leads to undesired effects, such as violating effect rules, it will receive a penalty. However, note that our model only specifies positive critical actions and does not explicitly identify actions that have possible negative consequences. Therefore, the implementation of a penalty depends on the specific domain.

## 6 EXPERIMENTS

We evaluate our framework and provide ablation studies in this section. Section 6.1 lists the algorithms we use for comparison. Section 6.2 provides the description of the environments and tasks. Section 6.3 presents the results of training efficiency and performance. Section 6.4 demonstrates the generalizability in different levels.

## 6.1 BASELINES

We extensively compare our framework to various DRL algorithms (DQN and PPO) learning from *rewards*, imitation learning methods (BC and GAIL) learning from *demonstrations*, advanced approaches (DQN-RBS and BC-PPO) that leverage both *rewards* and *demonstrations*, and an exploration method (RIDE) that maximizes *intrinsic and extrinsic rewards*.

- **DQN** (Mnih *et al.*, 2015) is an off-policy deep Q-learning algorithm.
- **PPO** (Schulman *et al.*, 2017) is a state-of-the-art on-policy DRL algorithm.

- **Behavior cloning** (**BC**; Ross *et al.*, 2011) imitates an expert by learning from demonstrations in a supervised manner.
- **Generative adversarial imitation learning** (**GAIL**; Ho & Ermon, 2016) mimics expert behaviors via learning a generative adversarial network whose generator is a policy.
- **DQN-RBS** initializes the replay buffer of DQN with demonstrations, allowing for a performance boost. This is inspired by the replay buffer spiking technique (Lipton *et al.*, 2016).
- **BC-PPO** pre-trains a BC policy using demonstrations and then fine-tunes the policy with PPO using rewards, similar to Video PreTraining (VPT; Baker *et al.*, 2022).
- **Rewarding impact-driven exploration** (**RIDE**; Raileanu & Rocktäschel, 2020) is an RL exploration method inspired by the intrinsic curiosity module (Pathak *et al.*, 2017).

## 6.2 ENVIRONMENTS & TASKS

To evaluate the proposed framework and the baselines, we design three groups of tasks in a $8 \times 8$ gridworld environment, where an agent can move along four directions $\{\texttt{up}, \texttt{down}, \texttt{left}, \texttt{right}\}$ and interact with objects. The tasks are described as follows. See Appendix C for more details.

SWITCH requires an agent to turn on the switches in sequential order. If it toggles the wrong switch, the progress will regress, which makes it challenging for DRL agents to solve the task through solely exploration. We design four tasks 4-SWITCH, 8-SWITCH, 4-DISTRACTORS and 4-ROOMS, where 4-SWITCH and 8-SWITCH evaluate the performance between different difficulties of tasks. 4-DISTRACTORS consist of four target switches and four distractor switches, and 4-ROOMS combines the configuration Minigrid *four-rooms* tasks (Chevalier-Boisvert *et al.*, 2023).

DOORKEY features a hierarchical task similar to the Minigrid *door-key* tasks, where the agent needs to open a door with a key and turn on a switch behind the door.

MINECRAFT is inspired by the computer game Minecraft and is similar to the environment in previous works (Sohn *et al.*, 2018; Andreas *et al.*, 2017; Sun *et al.*, 2020; Brooks *et al.*, 2021). The environment is designed for evaluation using multiple-task demonstrations. We select a simple task IRON, a difficult task ENHANCETABLE, and a multiple-goal task MULTIPLE.

For the methods that require demonstrations, we collect 20 demonstrations from corresponding tasks in SWITCH and DOORKEY and collect 64 multiple-task demonstrations from MULTIPLE for all tasks in MINECRAFT.

## 6.3 RESULTS

The experimental results in Figure 4 show that our framework outperforms all the baselines on challenging tasks (*e.g.*, 8-SWITCH, 4-DISTRACITORS, ENHANCETABLE, MULTIPLE) and performs competitively on simpler tasks (*e.g.*, DOORKEY, 4-SWITCHES, IRON). The imitation learning approaches, BC and GAIL, fail to learn all the tasks due to insufficient demonstrations and lack of exploration, while RIDE, BC-PPO, and DQN-RBS, which consider rewards online, fail on advanced tasks that require long-term planning. In contrast, our framework can leverage the knowledge from the same number of demonstrations and efficiently explore the environment, especially on the tasks 8-SWITCHES where all the baselines completely fail, as shown in Figure 4b. Moreover, our proposed framework is the most sample-efficient method in learning the DOORKEY task.

## 6.4 GENERALIZABILITY

Our framework employs the critical action model to achieve task-level generalizability, enabling the construction of novel task structures based on familiar critical actions. Additionally, we introduce genetic programming, renowned for its adaptability in reasoning symbolic rules as task substructures, thereby enhancing generalizability at the rule level. To define the domain gap, we denote the *original domain* as the domain where demonstrations are collected and the *variant domain* as the domain where agents learn. For rule-level generalizability, we define a variant critical action $\phi$ from $\psi$ where $\mathcal{V}(\textit{eff}(\phi)) = \mathcal{V}(\textit{eff}(\psi))$ and $\mathcal{V}(\textit{pre}(\phi)) = \mathcal{V}(\textit{pre}(\psi))$ while $\textit{eff}(\phi) \neq \textit{eff}(\psi)$ or $\textit{pre}(\phi) \neq \textit{pre}(\psi)$. If a critical action $\psi$ varies, the induced symbolic programs and the population can evolve and adapt to new substructures. Since $\mathcal{V}(\textit{pre}(\phi))$ and $\mathcal{V}(\textit{eff}(\phi))$ are known, the procedure starts from inducing effect rules $\textit{eff}(\phi) \neq \textit{eff}(\psi)$. Thus, the proposed framework can potentially achieve task generalization.

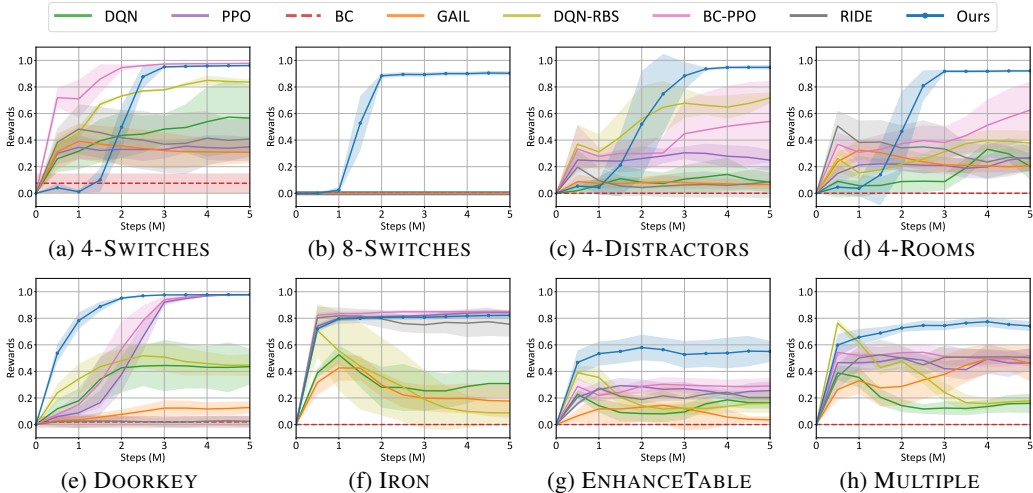

Figure 4: **Task performance.** We report the mean (line) and the standard deviation (shaded regions) of the training curves over 5M steps out of three runs. Our approach outperforms other methods, especially in advanced tasks.

**Setup.** To evaluate the generalizability of our framework and baselines, we consider 4-SWITCHES-(N+1) as the original domain and its variant domains, 8-SWITCHES-(N+1), 4-DISTRACTORS-(N+1), and 4-DISTRACTORS-(2N+1). In 8-SWITCHES-(N+1), we extend the number of switches from 4 to 8 to evaluate the generalization of task structures. In 4-DISTRACTORS-(N+1), 4 distractor switches are added to 4-SWITCHES-(N+1), and in 4-DISTRACTORS-(2N+1), the order of switches changes to $2n + 1$ (*e.g.*, $1 \rightarrow 3 \rightarrow 5 \rightarrow 7$), while the order of switches in N-DISTRACTORS-(N+1) is $n + 1$ (*e.g.*, $1 \rightarrow 2 \rightarrow 3 \rightarrow 4$). This series of settings evaluates if a method can generalize to different effect rules. We collect 200 demonstrations in 4-SWITCHES-(N+1) and and run 5M steps for all methods. For 4-DISTRACTORS-(2N+1), we collect *only* 4 additional demonstrations for all methods, and our framework leverages the previous populations of genetic programming to re-induce the rules, which only require few-shot demonstrations.

**Baselines.** We compare our framework with the best-performing baselines (BC-PPO) and the most widely used baseline (GAIL) for the generalization experiments.

**Results.** The results in Table 1 demonstrate that the performance of GAIL and BC-PPO drops in the variant domains, whereas our framework is able to generalize, highlighting its ability to construct novel rules and structures in the variant domains.

Table 1: **Generalization performance** in the original domain 4-SWITCHES and its variant domains.

| Task | GAIL | BC-PPO | Ours |
|---|---|---|---|
| 4-SWITCHES-(N+1) | 30%±8% | **97**%±0% | **96**%±1% |
| 8-SWITCHES-(N+1) | 10%±2% | 00%±0% | **90**%±2% |
| 4-DISTRACTORS-(N+1) | 10%±7% | 41%±4% | **95**%±2% |
| 4-DISTRACTORS-(2N+1) | 11%±6% | 33%±2% | **95**%±1% |

## 6.5 ABLATION STUDY

This section presents the ablation studies of the induction modules. Section 6.5.1 qualitatively examines the mutual information and Section 6.5.2 shows the accuracy of symbolic regression using genetic programming.

### 6.5.1 ACTION-EFFECT LINKAGE

In Section 5.1, we introduce action-effect linkages to discover the co-occurred effect variables and actions. Figure 5 presents the experimental results in MINECRAFT and shows the relationship between the logarithm of mutual information and action-effect linkages. The

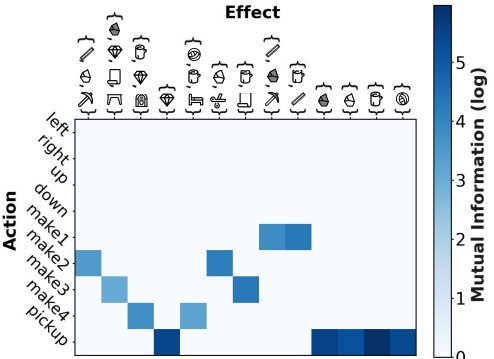

Figure 5: **Action-effect mutual information in MINECRAFT inferred by our framework.**

heat map visualizes the values of all action-effect pairs, with darker colors indicating higher val-

ues and stronger associations, highlighting the linkages. For instance, {📦 wood, ✏ stick} is the effect variables of make_stick as mentioned in Figure 1a and 1b, discovered by our framework from executing make1.

### 6.5.2 SYMBOLIC REGRESSION

The proposed framework necessitates a robust symbolic regression module to generate the symbolic rules. In Section 5.1, we introduce genetic programming as symbolic regression for induction. Since genetic programming is a randomized search method, empirical results are shown to discuss the success rate of finding correct rules and how much demonstrations are required to capture the symbolic rules.

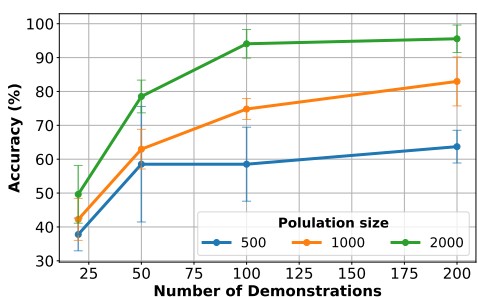

Figure 6: **Symbolic regression accuracy.**

The experiment setting is described as follows. In MINECRAFT environment, there are 27 effect rules listed in Table 3. We sample different numbers of demonstrations from random subtasks, and the number of population are 500, 1000, and 2000. Other parameters of genetic programming are the same as the setting in Table 2. We calculate the number of programs which is equivalent to the ground truth after simplification. The result is the average accuracy out of five runs shown in Figure 6. We claim that the effect rules can be induced via genetic programming when a sufficient number of demonstrations and programs in the population are available. Noting that the results are related to the diversity of the data. In theoretically, each $n$-polynomial rule requires more than $n + 1$ points for regression. In addition, critical action networks can still be built when some rules are inequivalent to the ground truth due to the bias of the data, as long as the rules match with the precondition of succeeding critical actions.

## 7 DISCUSSION

We present a framework to address sparse-reward, goal-directed MDP tasks by integrating DRL and classical planning techniques. Our proposed framework represents symbolic knowledge as critical actions and employs a procedure to automatically extract knowledge from demonstrations. This combination of inductive learning (*i.e.*, DRL) and deductive learning (*i.e.*, classical planning) enables our framework to perform explicit high-level planning and accurate low-level execution, allowing for robust task performance and generalizing to unseen domains. Additionally, evolutionary computation provides adaptability at the rule level by inducing the task substructures.

Specifically, by representing knowledge as critical actions and employing critical action networks, we provide a structured and organized mechanism for capturing and utilizing symbolic knowledge within DRL. The proposed procedures of subtask decomposition combine planning and DRL, leading to effective and efficient learning in goal-directed tasks. Furthermore, the compositionality of the critical action model allows for different levels of generalization, highlighting its potential to address a wide range of general problems. In sum, our work offers a holistic perspective to effectively handle general goal-directed decision-making problems with the integration of inductive and deductive learning.

This work extensively evaluates our proposed framework on deterministic, fully observable environments within the integer domain. To extend our proposed framework to complex domains, such as continuous input, we can potentially leverage recent studies that have developed genetic programming for different variable types (Virgolin *et al.*, 2017; 2021). When facing stochastic or partially observable environments, the induction module can fail to induce some critical actions because of the uncertainty. In a worst-case scenario, when the induction module produces no rule, the proposed framework simply reduces to the backbone DRL algorithm (*i.e.*, PPO) without any intrinsic reward. One potential direction to address this issue is to extend the induction module to recognize and infer probabilistic rules (Pasula *et al.*, 2007; Arora *et al.*, 2018), which is left for future work.

## 8 ACKNOWLEDGEMENT

The authors would like to thank the support of the National Science and Technology Council in Taiwan under grant No. NSTC 111-2221-E-002-189 and National Taiwan University under grant No. 112L891103. Shao-Hua Sun was partially supported by the Yushan Fellow Program by the Ministry of Education, Taiwan.

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

APPENDIX

# A  PRELIMINARY DEFINITION

This section provides the annotations of the Markov decision process (MDP) and planning specification discussed in Sectino 4. Section A.1 gives the formulation of MDP problems and Section A.2 explains the definition and the annotation of planning domain definition language (PDDL).

## A.1  MARKOV DECISION PROCESS

A decision-making problem is formulated as a Markov decision process (MDP). MDP consists of a five-tuple $\langle \mathcal{S}, \mathcal{A}, T, R, \gamma \rangle$, where $\mathcal{S}$ denotes state space, $\mathcal{A}$ denotes action space, $T : \mathcal{S} \times \mathcal{A} \rightarrow \mathcal{S}$ denotes a transition function, $R : \mathcal{S} \times \mathcal{A} \rightarrow \mathbb{R}$ denotes a reward function, and $\gamma \in (0, 1]$ denotes a discounting factor.

In contrast with classical planning, problems in reinforcement learning represent the actions and the states with vectors of numeric values instead of literal conjunctions. A state in $\mathcal{S}$ is a $n$-dimension vector $s$, where each entry $s_i$, $i \in \{1, 2, ..., n\}$, represents the value of a numeric variable. In this work, we focus on goal-directed sparse-reward problems. That is, given an initial state, the objective is to find a policy to reach a desired goal state, and the agent only receives rewards when reaching the goal state.

## A.2  PLANNING DOMAIN DEFINITION LANGUAGE

PDDL is a language in first-order logic to describe the domains and the problems. A domain description includes the specification of objects, variables, and action models with preconditions and effects. A problem description includes an initial state and goal specification. These standard language specifications allow off-the-shelf planners to deduce the optimal action sequence of the goal.

For theory formalism, we follow the representation of SAS$^+$. To distinguish with MDP state space $\mathcal{S}$, the state space in the planning domain are denoted as $\mathcal{S}'$. A SAS$^+$ task $\Pi$ can be represented as a tuple $\langle \mathcal{V}, \mathcal{O}, s'_{init}, p'_{goal} \rangle$, where $\mathcal{V}$ is a set of variables, and $\mathcal{O}$ is a set of operators in the domain. A state $s' \in \mathcal{S}'$ in the planning domain is an assignment to $\mathcal{V}$, and $s'_v \in \mathbb{R}$ is the value assigned to the variable $v \in \mathcal{V}$ in $s'$. $p' \subset s'$ is a partial state of $s'$, where $p'$ is the assignment to $\mathcal{V}(p') \subset \mathcal{V}$. Specifically, $s'_{init} \in \mathcal{S}'$ is the initial state, and $p'_{goal}$ is a partial state of the goal specification.

A logical condition $l_v$ describes the relation between the variable $v$ and an distinct value (*e.g.,* wood = 1, stick $\geq$ 2). Each operator $o \in \mathcal{O}$ can be described as an action schema in a pair $\langle pre(o), \mathit{eff}(o) \rangle$, where $pre(o)$ is a conjunction of logical conditions denoted the precondition needed to be satisfied before executing $o$, $\mathit{eff}(o)$ is a set of functions denoted the change toward the state variables after executing $o$. The prevail condition is a subset of $pre(o)$ which holds during the action and does not affect by the effect, denoted as $prv(o) = \{l_v \mid l_v \in pre(o), v \notin \mathcal{V}(\mathit{eff}(o))\}$. An operator $o$ is admissible in state $s'$ iff $pre(o) \subset s'$ and $prv(o) \subset s'$.

# B  IMPLEMENTATION DETAIL

In this section, we elaborate on the implementation detail of the proposed framework that was used in the experiments. We implement several modules using off-the-shelf packages and approaches, including genetic programming as symbolic regression with *gplearn*, the agents with PPO, and the intrinsic reward function we use in the experiments.

## B.1  ALGORITHM IN EXTRACTING ACTION-EFFECT LINKAGES

In Algorithm 1, for each MDP action, we calculate the mutual information between the action and all combinations of effect variables in demonstrations. Then, we apply the two-center clustering method to determine the threshold shown in Algorithm 2. Two-center clustering finds a threshold value that separates a given data into two clusters, which minimizes the sum of distances of data points from their respective cluster centers. We take the logarithm of mutual information as the metric to avoid incorrect thresholds caused by extremely high mutual information.

---

**Algorithm 1** Extracting Action-Effect Linkages

---

**Input:** Demonstrations, action set $\mathcal{A}$, effect set $\mathcal{E}$
**Output:** Action-effect pairs with linkages $L$
    $L \leftarrow \emptyset$
    **for** $a$ in $\mathcal{A}$ **do**
        $N_a \leftarrow \emptyset$
        **for** $\mathcal{V}(\mathit{eff}(\cdot))$ in $\mathcal{E}$ **do**
            $N_a \leftarrow N_a \cup M_{(a,\mathcal{V}(\mathit{eff}(\cdot)))}$
        **end for**
        $t \leftarrow$ Two-Center-Clustering$(N_a)$
        **for** $\mathcal{V}(\mathit{eff}(\cdot))$ in $\mathcal{E}$ **do**
            **if** $M_{(a,\mathcal{V}(\mathit{eff}(\cdot)))} \geq t$ **then**
                $L \leftarrow L \cup (a, \mathcal{V}(\mathit{eff}(\cdot)))$
            **end if**
        **end for**
    **end for**

---

**Algorithm 2** Two-Center-Clustering

---

**Input:** Data $D$ with length $n$
**Output:** Threshold of two clusters
    $cluster_1, cluster_2 \leftarrow \emptyset$
    $c_1, c_2 \leftarrow min(D), max(D)$
    **if** $c_1 = c_2$ **then**
        **return** $c_1$
    **end if**
    $terminated \leftarrow False$
    **while** not $terminated$ **do**:
        **for** $i = 1$ to $n$ **do**
            **if** $|D[i] - c_1| < |D[i] - c_2|$ **then**
                $cluster_1 \leftarrow D[i]$
            **else**
                $cluster_2 \leftarrow D[i]$
            **end if**
        **end for**
        $c_1', c_2' \leftarrow mean(cluster_1), mean(cluster_2)$
        **if** $c_1 = c_1'$ and $c_2 = c_2'$ **then**
            $terminated \leftarrow True$
        **end if**
        $c_1, c_2 \leftarrow c_1', c_2'$
    **end while**
    **return** $(c_1 + c_2)/2$

---

### B.2 GENETIC PROGRAMMING

Genetic programming is employed as a symbolic regressor for determining symbolic effect rules in the proposed methods, illustrated in Figure 3a. We use the *gplearn* package for implementation and the parameter settings of are shown in Table 2. Given the action-effect linkage $(a, \mathcal{V}(\mathit{eff}(\cdot)))$, the transitions with action $a$ are selected as the training data. For each effect variable $v$ in $\mathcal{V}(\mathit{eff}(\cdot))$, the algorithm's objective is to find the program $\mathit{eff}(\cdot)_v$ that predicts $v$ after executing the action with the highest accuracy.

Each program is represented as an expression tree where input is the current state in the transition and output is the predicted value. The algorithm comprises several steps: initialization, evaluation, selection, crossover, and mutation. Initially, the population, which is a set of programs, is randomly generated. Fitness evaluation is then performed on all programs; a subset of programs with the highest fitness values is selected. These programs serve as parents to produce offspring through crossover and mutation mechanisms. Through iterative selection and production, the evolution of

Table 2: **The parameter setting of *gplearn*.** The parameter with two values indicates that the settings are different in two phases.

| Parameters | Value (first/second phase) |
|---|---|
| population_size | 2000/2000 |
| tournament_size | 40/40 |
| generations | 20/10 |
| p_crossover | 0.6/0.6 |
| p_subtree_mutation | 0.2/0.2 |
| p_hoist_mutation | 0.1/0.1 |
| p_point_mutation | 0.05/0.05 |
| max_samples | 0.95/0.95 |
| init_depth | (2,6)/(2,6) |
| parsimony_coefficient | 0.0001/0.005 |
| function_set | $\{+, -, \times, \div, \text{inc}, \text{dec}\}/\{+, -, \times, \div, \text{inc}, \text{dec}\}$ |

the population to discover the programs that best fit the given data. The evaluation metric used in genetic programming is the percentage of correct effect prediction shown below:

$$fitness(eff(\cdot)_v) = \frac{\text{\# of transitions with } (a, \mathcal{V}(eff(\cdot))) \text{ consistent with } eff(\cdot)_v}{\text{\# of transitions with } (a, \mathcal{V}(eff(\cdot)))}, \qquad (2)$$

where a transition consistent with $eff(\cdot)_v$ means that the predicted effect value $eff(\cdot)_v(s_v)$ is consistent with the actual one $T(s, a)$ given the transition $\langle s, a, T(s, a) \rangle$. To prevent bloat issues in which the program grows extremely large to fit the data, the algorithm contains two phases: exploring and pruning. The best programs with the highest accuracy are determined in the exploring phase. Subsequently, in the pruning phase, we set high parsimony to prune the program.

### B.3 Decision Tree Method

The proposed framework uses classification and regression tree (CART) (Loh, 2011) to build decision trees. CART is a supervised learning algorithm that generates binary trees by recursive partitioning, where each internal node represents a decision based on a specific variable, and each leaf node represents a prediction. Let data partitioned at the internal node $m$ denoted as $D_m$ with $n_m$ samples. The algorithm aims to find a decision with a variable $q$ and a threshold $t$ to partition $D_m$ into two subsets $D_m^0$ and $D_m^1$ with $n_m^0$ and $n_m^1$ samples. The loss function of the partition is defined as follows:

$$G(D_m, q, t) = \frac{n_m^0}{n_m} H(D_m^0) + \frac{n_m^1}{n_m} H(D_m^1), \qquad (3)$$

where $H(D_m^i)$ is the entropy of $D_m^i$. In each partition, the algorithm's objective is to find the $(q, t)$ that minimizes $G(D_m, q, t)$ at node $m$. This process is repeated until a stopping criterion is met.

In the given transition $\langle s, a, T(s, a) \rangle$ with $a$ in demonstrations, the current states $s$ are taken as inputs to a decision tree, and the outcome of the decision tree is a true value that whether $T(s, a)$ consistent with the rules in $eff(\cdot)$. After generating a decision tree by CART, the model of this decision tree is then transferred into a conjunction of rules by logical simplification and set as the precondition rules $pre(\cdot)$, while $\mathcal{V}(pre(\cdot))$ is the set of variables mentioned in $pre(\cdot)$. Considering the precondition is the conjunction of the precondition rules while the formula of the decision tree may involve disjunction, the decision tree model is transferred into the disjunctive normal form. Each clause in the disjunctive normal form is considered the precondition for different critical actions.

### B.4 Deep Reinforcement Learning Module

In our research, we adopt the Proximal Policy Optimization (PPO) as the foundation for our framework and the baseline in this work, utilizing the *torch-ac* library (Willems, 2022) for implementation. The architecture of our model is designed to encode an 8x8 gridworld and state information.

The gridworld is handled by a convolutional neural network (CNN) comprising four layers. The network effectively processes the input images, which are formatted in a three-channel setup, representing the object, color, and status respectively. The first convolutional layer utilizes 32 filters with a kernel size of 3×3 and a stride of 2×2. Subsequent layers employ 2×2 kernels, with channel sizes incrementing through 64, 96, to 128. Each layer integrates the ReLU activation function.

Parallelly, the state representation, converted into PDDL, is encoded through a two-layer fully-connected network. Each of these layers contains 64 neurons and leverages the ReLU activation function for non-linear transformation.

Two types of encoded observation are concatenated and encoded by another two-layer $64 \times 64$ fully-connected network. The output-encoded observation is then used as the input of the actor network and the critic network. Both the encoding networks and the actor-critic networks are trained.

### B.5 REWARD FUNCTION

During the training stage, we train an agent with intrinsic rewards generated from the critical action network. The modified rewards function is illustrated as follows:

$$R_{int}(s) = \begin{cases} +1 & \text{if execute a critical action,} \\ 0 & \text{otherwise.} \end{cases} \tag{4}$$

Given the original reward function $R$ as extrinsic rewards, the overall reward of the MDP problem is $R_{mod}(s) = R(s) + R_{int}(s)$. In the experiment, the reward function is defined as

$$R = \begin{cases} \frac{step_{max} - step}{step_{max}} & \text{if the episode terminated,} \\ 0 & \text{otherwise,} \end{cases} \tag{5}$$

where $step_{max}$ is the maximum number of steps in the environment, and $step$ is where $step$ is the current number of steps the agent has done. The setting of the maximum number of steps in each environment is described in Appendix C.

## C TEST ENVIRONMENTS

We use three MDP environments: SWITCH, DOORKEY, and MINECRAFT for evaluation. The first environment SWITCH tests the ability to achieve sequential tasks. The second environment DOORKEY is similar to *door-key* in *Minigrid* which is a baseline environment with hierarchical tasks. The third environment MINECRAFT is designed to evaluate the ability to construct various task structures with multiple subtasks for compositional tasks. The following sections provide a description of the environments. The maximum number of steps in DOORKEY is 1600, while in SWITCH and MINECRAFT is 25600.

### C.1 SWITCH

The environment SWITCH is designed to evaluate the ability to solve hierarchical tasks. In SWITCH, several switches are placed on the grid. The objective of the agent in SWITCH is to sequentially turn on switches in a pre-determined order.

We define the state variables as $\mathcal{V} = \{\texttt{at\_switch}, \texttt{next\_switch}, \texttt{goal\_switch}\}$. $\texttt{at\_switch}$ indicates the switch the agent stays at. If the agent does not stay at any switch, this variable is set to zero. $\texttt{next\_switch}$ indicates which switch should be activated in the following actions. $\texttt{goal\_switch}$ denotes the last switch and also implies how many switches should be turned on. The action space $\mathcal{A}$ contains five actions: $\{\texttt{left}, \texttt{right}, \texttt{up}, \texttt{down}, \texttt{toggle}\}$. $\texttt{toggle}$ enables the agent to activate or deactivate a switch.

The switches have three states, including *available*, *on* and *off*. The agent can turn the *available* switch to *on*. If the agent executes the action $\texttt{toggle}$ at the switch, it will be deactivated and turned to *available*. The agent can not change the status of the *off* switch until the predecessor switch is *on*. If a *on* switch is turned to *available*, all subsequent switches will also be deactivated. This makes it challenging for RL agents to solve the task through random walks or exploration alone.

For various evaluations, we design several situations, including the number of switches, sequential order, and distractors. In the following sections, the settings of the tasks are listed and the illustrations can be found in Figure 7.

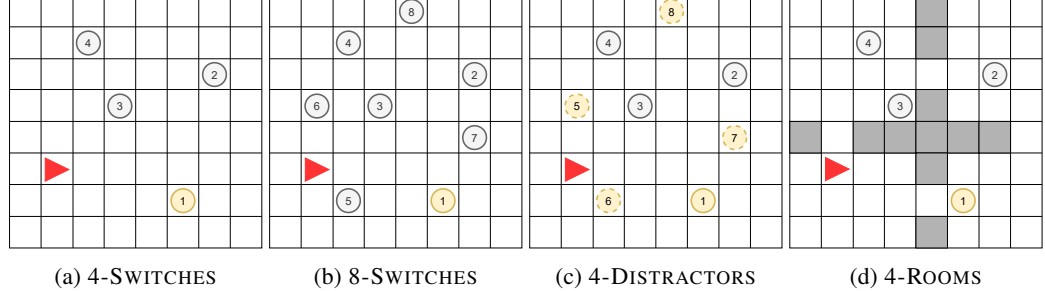

|  |  |  |  |
|---|---|---|---|
| (a) 4-SWITCHES | (b) 8-SWITCHES | (c) 4-DISTRACTORS | (d) 4-ROOMS |

Figure 7: **Visualization of SWITCH environments.** N-SWITCHES in (a) and (b) shows the tasks which consist of 4 and 8 switches in incremental order. (c) N-DISTRACTORS is the variant N-SWITHES with other $n$ distractor switches (the circle with dotted line). (d) combines *four-room* environment in *Minigird*, testing the navigation ability of the agent.

- **N-SWITCHES.** When the number of switches increases, the tasks become more difficult as it has more chance to turn off the switch. This setting evaluates the performance of different difficulties of tasks.

- **N-DISTRACTORS.** *Available* switches are added as distractors to the environment. The agent can turn on and off the distractor switches, but it does not help to achieve the tasks. In $n$-switch incremental-order tasks, the switches labeled $n + 1$ to $2n$ are set as distractors, while in $n$-switch incremental-order tasks, the switches labeled $2, 4, ..., 2n$ are set as distractors. This setting evaluates whether the agent acquires the ability to select correct switches and neglects the incorrect ones.

- **4-ROOMS.** Two lines of walls divide the gridworld into four rooms according to the four-room configuration in *Minigrid*. Every two rooms are interconnected by a gap in the walls. In this scenario, the agent must navigate through the rooms considering the walls to activate the switches. This setting evaluates the efficacy of DRL involving navigating obstacles at low-level execution.

- **Order of the switches.** To evaluate generalizability, we define two types of orders including (N+1) and (2N+1). (N+1) indicates that the switches should be turned on in incremental order (*i.e.*, $1 \to 2 \to 3 \to 4...$) where (goal_switch = n), and (2N+1) indicates that the switches should be turned on in odd order (*i.e.*, $1 \to 3 \to 5 \to 7...$) where (goal_switch = 2n + 1). The order are labeled after the task name (*e.g.*, 4-DISTRACTORS-(N+1)), and by default the order is (N+1).

## C.2 DOORKEY

The environment DOORKEY presents a task where an agent must collect a key to unlock a door and turn on the switch behind the door. It is a basic setting which can be used to evaluate the ability to solve hierarchical tasks.

We define the state variables as $\mathcal{V} = \{$agent_dir, has_key, door_state, at_switch, at_door, next_switch$\}$. agent_dir indicates the agent direction, has_key indicates whether the agent holds the key, and door_state indicates the state of the door, including open, closed, and locked. The action space $\mathcal{A}$ contains seven actions: $\{$left, right, up, down, toggle, pickup, drop$\}$. toggle enables the agent to activate a switch or open the door.

## C.3 MINECRAFT

MINECRAFT is inspired by the computer game *Minecraft* and is similar to the environment in previous works (Sohn *et al.*, 2018; Andreas *et al.*, 2017; Sun *et al.*, 2020; Brooks *et al.*, 2021) illustrated in Figure 8a. The agent can pick up the primary materials on the map and make different tools in specific places consuming the materials. The goal of each task is to acquire the desired materials or tools. The state variables include $\{$x, y, at_< place >, < inventory >$\}$, where at_< place > denotes whether the agent is at the place, and $<$ inventory $>$ denotes the number of materials or tools the agent holds.

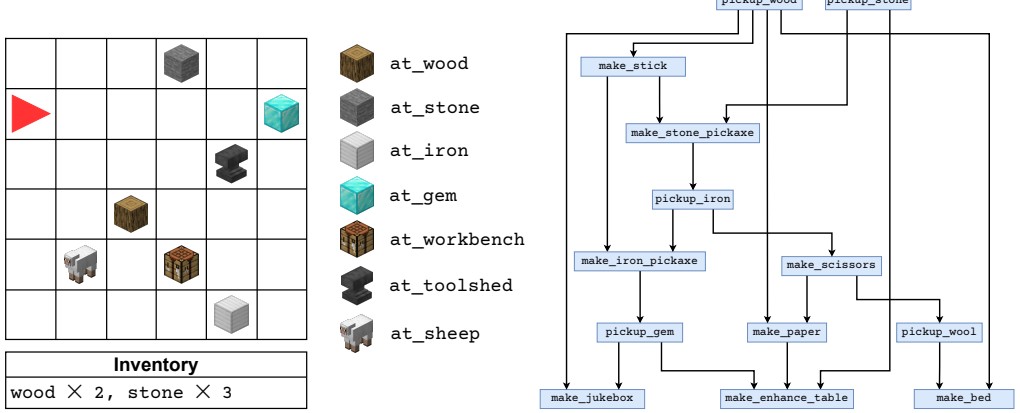

(a) Visualization of MINECRAFT environment.    (b) Dependency of subtasks in MINECRAFT.

Figure 8: **Illustration of configuration and subtask dependencies in MINECRAFT environment.**
(a) There are 7 locations in MINECRAFT environment, and the agent needs to collect materials and craft tools at specific location. (b) illustrate the dependencies of the critical actions. For instance, to execute pickup_iron, the agent needs to make a stone pickaxe. The graph ignores preconditions, effects, and the number of executions required.

In our experiments, thirteen types of items are designed in the inventory: wood, stone, stick, iron, gen, stone_pickaxe, iron_pickaxe, wool, paper, scissors, bed, jukebox, enhance_table, and there are seven places on the gird world: at_wood, at_stone, at_iron, at_gem, at_sheep, at_workbench, at_toolshed.

The action space $\mathcal{A}$ contains eight actions: {left, right, up, down, make1, make2, make3, make4}. The agent crafts different items when executing different make actions (make1, make2, make3, make4) and at different places (workbench or toolshed). The formulas of the items are listed in Table 3, and the dependency of the subtasks is illustrated in Figure 8b. The agent needs to get the materials to create desired items. We test two single tasks with different difficulties IRON and ENHANCETABLE, and a multiple task MULTIPLE that sample the goal at random.

Table 3: Formulas in MINECRAFT environment. The MDP actions, precondtions and effects are the same as their corresponding critical actions.

| Inventory | MDP Action | Preconditions Effects | | | |
|---|---|---|---|---|---|
| wood | pickup | $\texttt{at\_wood} = 1$ $\texttt{wood} + 1$ | | | |
| stone | pickup | $\texttt{at\_stone} = 1$ $\texttt{stone} + 1$ | | | |
| iron | pickup | $\texttt{at\_iron} = 1$ $\texttt{iron} + 1$ | $\texttt{stone\_pickaxe} \geq 1$ | | |
| gem | pickup | $\texttt{at\_gem} = 1$ $\texttt{gem} + 1$ | $\texttt{iron\_pickaxe} \geq 1$ | | |
| wool | pickup | $\texttt{at\_sheep} = 1$ $\texttt{wool} + 1$ | $\texttt{scissors} \geq 1$ | | |
| stick | make1 | $\texttt{at\_workbench} = 1$ $\texttt{stick} + 1$ | $\texttt{wood} - 1$ | | |
| stone_pickaxe | make1 | $\texttt{at\_toolshed} = 1$ $\texttt{stone\_pickaxe} + 1$ | $\texttt{stone} - 3$ | $\texttt{stick} - 2$ | |
| iron_pickaxe | make2 | $\texttt{at\_toolshed} = 1$ $\texttt{iron\_pickaxe} + 1$ | $\texttt{iron} - 3$ | $\texttt{stick} - 2$ | |
| iscissors | make2 | $\texttt{at\_workbench} = 1$ $\texttt{scissors} + 1$ | $\texttt{iron} - 2$ | | |
| paper | make3 | $\texttt{at\_workbench} = 1$ $\texttt{paper} - 1$ | $\texttt{scissors} \geq 1$ $\texttt{wood} - 1$ | | |
| bed | make3 | $\texttt{at\_toolshed} = 1$ $\texttt{bed} + 1$ | $\texttt{wood} - 3$ | $\texttt{wool} - 3$ | |
| jukebox | make4 | $\texttt{at\_workbench} = 1$ $\texttt{jukebox} + 1$ | $\texttt{wood} - 3$ | $\texttt{gem} - 1$ | |
| enhance_table | make4 | $\texttt{at\_toolshed} = 1$ $\texttt{enhance\_table} + 1$ | $\texttt{stone} - 1$ | $\texttt{paper} - 2$ | $\texttt{gem} - 1$ |

## D  CRITICAL ACTION

The critical action and its graph are shown in this section. Tables 4 and 5 show the critical actions in DOORKEY and SWITCH environments. The rules of critical actions in MINECRAFT environment are already listed in Table 3 and are omitted in this section. Figures 9, 10, 11 and 12 illustrate the critical action networks of N-SWITCHES, DOORKEY, IRON and ENHANCETABLE, respectively.

Table 4: Critical actions in SWITCH environment.

| Action Name | MDP Action | Preconditions Effects |
|---|---|---|
| turn_on | toggle | $\texttt{at\_switch} = \texttt{next\_switch}$ $\texttt{next\_switch} + 1$ |

Table 5: Critical actions in DOORKEY environment.

| Action Name | MDP Action | Preconditions Effects | | |
|---|---|---|---|---|
| pickup_key | pickup | $at\_key = 1$ $at\_key = 0$ | $has\_key = 1$ | |
| open_door | toggle | $door\_state = 0$ $door\_state = 2$ | $at\_door = 1$ | $agent\_dir = 1$ |
| turn_on | toggle | $at\_switch = 1$ $next\_switch + 1$ | $door\_state = 2$ | |

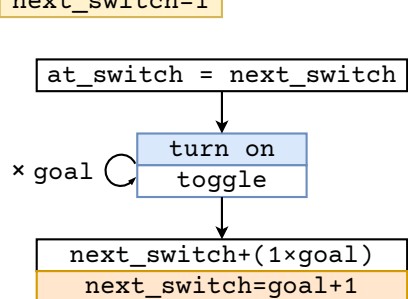

Figure 9: Critical action network of SWITCH

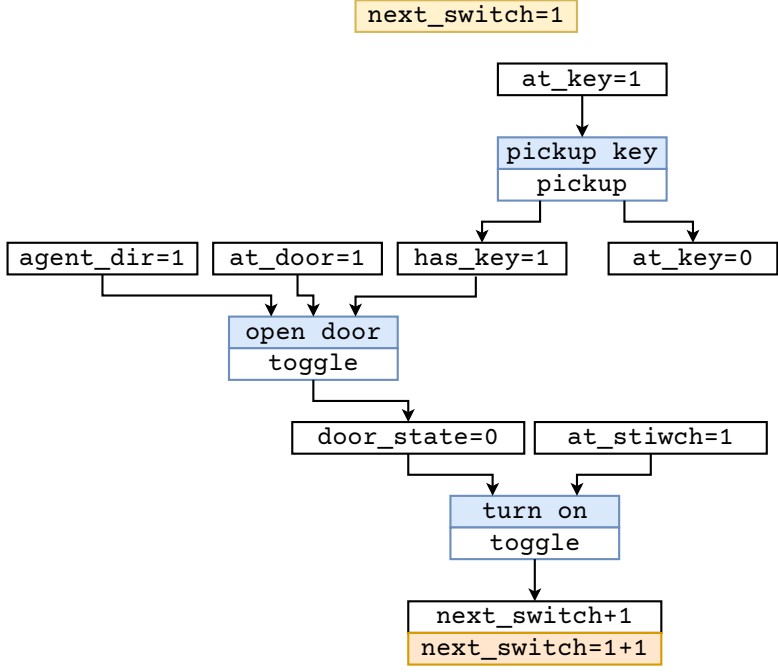

Figure 10: Critical action network of DOORKEY

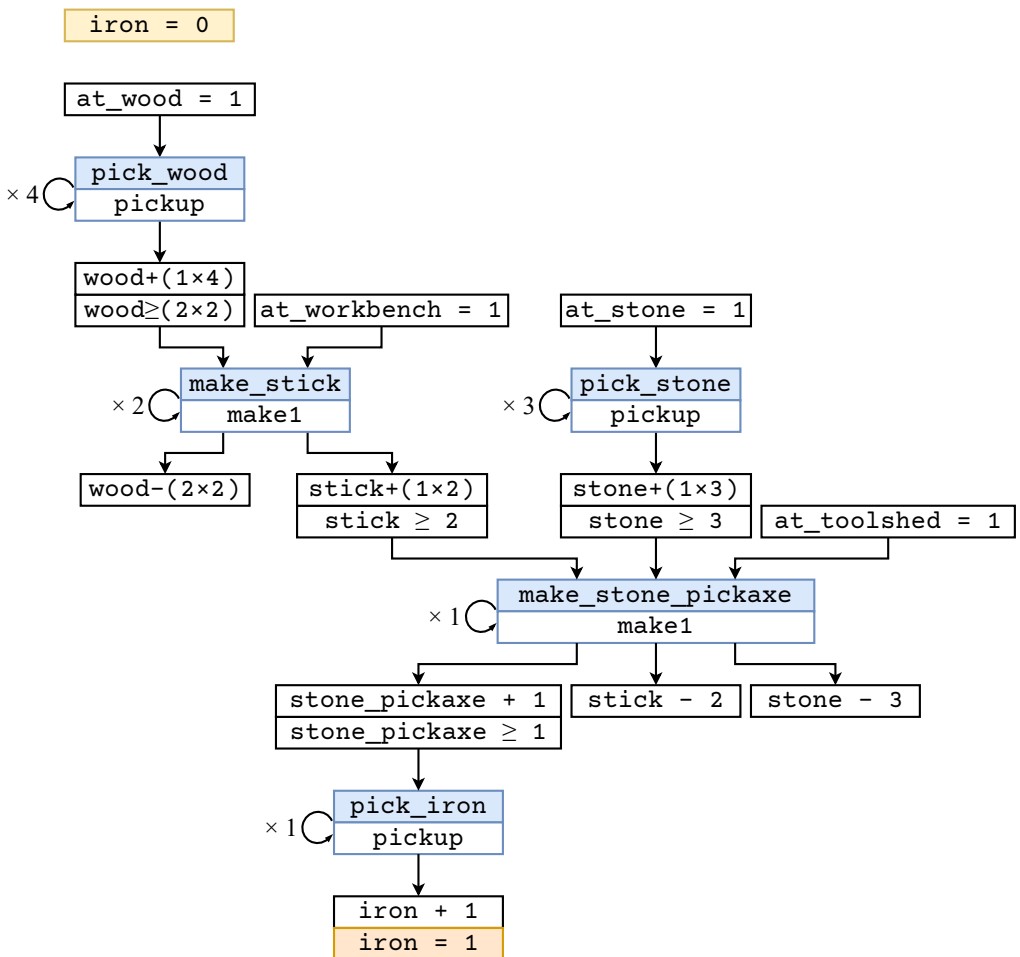

Figure 11: Critical action network of IRON

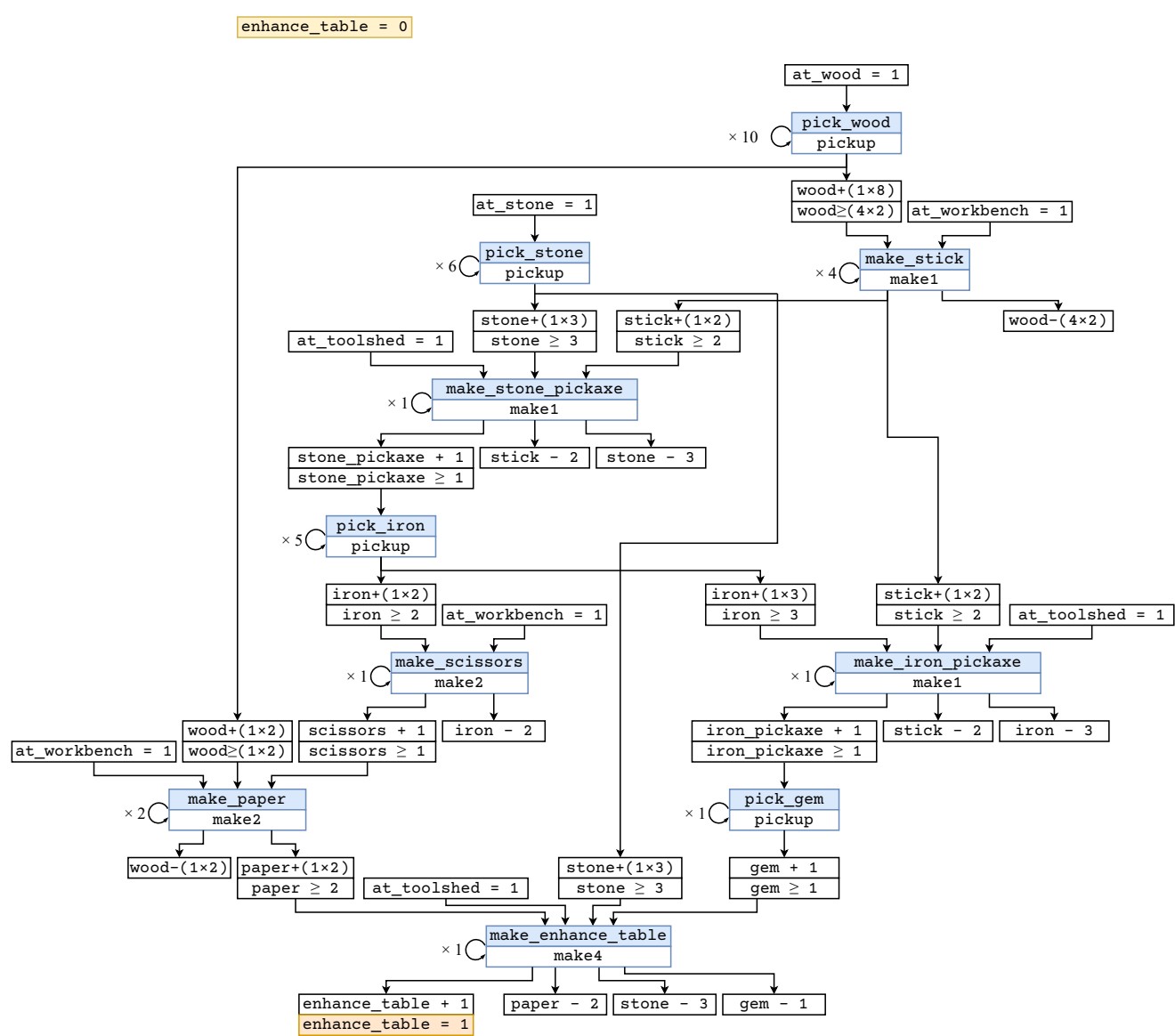

Figure 12: Critical action network of ENHANCETABLE

