# OpenReview forum: "Integrating Planning and Deep Reinforcement Learning via Automatic Induction of Task Substructures"
_ICLR.cc/2024/Conference — ICLR 2024 poster_

### Official Review · Reviewer_n4PR · 2023-10-31

**Soundness:** 3 good
**Presentation:** 3 good
**Contribution:** 3 good
**Rating:** 6
**Confidence:** 3

**Summary:**

In this paper, the authors seek to speed up deep reinforcement learning by inferring a classical planning model from expert demonstrations. Specifically, they infer PDDL-style preconditions and effects of "critical actions" (actions that are crucial for success and must be executed in a particular order) via genetic programming. Later, when training an RL agent to perform the granular actions required to complete the task, they use backward-chaining to infer the next critical action that should be taken, and provide an intrinsic reward to the RL for completing that action. Experiments in three gridworld domains show that this approach learns faster than several competing methods (vanilla RL, behavioural cloning, and an alternative intrinsic reward approach), and further show that the inferred knowledge can be readily applied to new task instances.

**Strengths:**

RL algorithms would have far more practical value if they could solve sparse-reward, goal-directed tasks quickly, since it's generally easier to specify tasks in these terms, especially for non-experts. Humans clearly do use some form of planning to solve such tasks, and intuitively we do learn high-level rules akin to action schemas, so this line of work is natural and well-motivated.

I'm familiar with much of the related work on hierarchical task learning, and while I'm not 100% sure about the authors' claim that "induc(ing) the required subtasks has not been addressed" (for example, Icarte et al., 2019 arguably do this to some degree), their approach to the problem is novel. I certainly haven't seen genetic programming applied in this context before, and it's an interesting idea!

**Weaknesses:**

The main issue I have with the paper is that the limitations/assumptions aren't sufficiently discussed. In particular:
- Deep reinforcement learning applications often involve learning from high dimensional input (e.g., pixels) or dealing with continuous controls (e.g., MuJoCo, real-word robotics). However, it's hard to see how the proposed approach would extend beyond discrete, gridworld-style tasks. At the top of page 6 it's claimed: "This choice of function set is consistent with the numerical variable representation commonly employed in DRL tasks", but seemingly the true assumption here is that the state can be encoded as a vector of *integers*. If true, this should be acknowledged, and either way I'd like to see a discussion of how the approach could be extended to more complex domains.
- The assumptions around "critical actions" aren't clear. It seems that the approach requires a human expert to deem which state variables are important (e.g., those corresponding to the inventory in Minecraft) and which are non-critical (e.g., the player's position). Again, this ought to be discussed.

A secondary concern is that there doesn't appear to be any source code included with the submission (correct me if I'm wrong) and the reinforcement learning setup isn't clear. For example:
- How is the 8x8 gridworld represented to the agent? Do you use different channels to represent different types of object, or just a single channel with different numerical values?
- Why is the agent paid a decaying reward based on the current step number (Appendix B.5)? Is this meant to be paid only at episode termination, or at every step? (The former would make more sense to me, but it's not expressed like this.)
- The gridworld and the PDDL state are encoded via a convolutional neural net and a fully-connected network, respectively. Are these encoders trained or fixed? What are kernel sizes of the CNN?
- I found the generalizability experiments quite confusing. Is the "ours" agent retrained in the variant domains after re-inducing the rules? (I can't see how it would generalize otherwise.) Are the original demonstrations for GAIL and BC-PPO discarded? Is BC-PPO retrained in the variant domain?

**Questions:**

Have you considered extending the approach to work without expert demonstrations? It seems to me that the induction module should still work, although it probably wouldn't be able to infer as many rules at the start of training (because certain types of transition wouldn't have been seen yet). However, rather than backward chaining from the goal, you could potentially backward chain from novel states, so as to encourage the agent to explore.

---

> ### Author Response · Authors · 2023-11-21
> **Response to Reviewer n4PR (1/2)**
>
> We sincerely thank the reviewer for the thorough and constructive comments. Please find the response to your questions below.
>
> > Have you considered extending the approach to work without expert demonstrations? It seems to me that the induction module should still work, although it probably wouldn't be able to infer as many rules at the start of training (because certain types of transition wouldn't have been seen yet).
>
> We have considered the approach that
>
> - induces incomplete critical actions from few-shot demonstrations and re-induces the critical actions using the data collected from the training agent, and
> - induces critical actions only with the data collected from the training agent.
>
> Our framework allows for learning with trajectories collected during training instead of demonstrations. To test the efficacy of our framework without demonstrations, we train a PPO agent from scratch and collect data online in the Minecraft MULTIPLE task. During training, we (re-)induce the action model every 40k steps, sampling 80k transition data for induction. We found that the main difference is the accuracy of genetic programming, which is listed as follows:
>
> | Steps | Accuracy |
> | --- | --- |
> | 40k | 47 $\pm$ 6% |
> | 80k | 49 $\pm$ 5% |
> | 120k | 49 $\pm$ 6% |
> | 160k | 48 $\pm$ 6% |
> | 200k | 49 $\pm$ 5% |
>
> The result shows that we can induce 50% of rules using the data collected by the agent learning from scratch. To be more specific, the rules in basic subtasks such as making a stick and picking up a wood are correct, while for advanced subtasks such as making an enhance table, GP overfits due to insufficient samples. This can be improved by adopting exploration-based methods (e.g., ICM or RND), which are left for our future work.
>
> > However, rather than backward chaining from the goal, you could potentially backward chain from novel states, so as to encourage the agent to explore.
>
> Thanks for the suggestion on backward chaining from novel states. It indeed addresses the issue of lacking critical actions of advanced subtasks which is required for backward chaining. Also, it will be more general if the agent can explore the environment intentionally without explicitly assuming the goal.
>
> > The main issue I have with the paper is that the limitations/assumptions aren't sufficiently discussed.
>
> We thank the reviewers for raising the issue of insufficient limitations and assumptions. We have discussed some of the limitations in Section 7 and some assumptions in Sections 3 and 4, and we are willing to discuss more sufficiently with the reviewer.
>
> Our framework focuses on deterministic, fully observable settings in the integer domain. Future works are needed to extend the method to a stochastic, partially observable, more complex domain. We note that our method is essentially a DRL agent, which performs equivalent to the standard DRL algorithm if the framework fails to learn any action model.
>
> > I'd like to see a discussion of how the approach could be extended to more complex domains.
>
> In our method, the variation of the framework for different domains is the model of the effect rules and precondition rules. There are more complex domains including continuous variables and high dimension input. For continuous input, genetic programming has developed several techniques focused on the symbolic regression of continuous functions, such as EllynGP and GP-GOMEA [1, 2]. The assumption and the method of threshold on continuous rules are required when inducing preconditions. Furthermore, GP can extend to various domains once the variable types are known. For instance, [3] focuses on symbolic regression using genetic programming on real-world benchmark datasets with continuous values, while [4] works on Boolean circuit benchmark problems.
>
> For image or high-dimensional input, several studies have adopted genetic programming to image input. The approach includes embedding images to vectors through CNN [5] or using Cartesian Genetic Programming, where the programs are represented in a grid of nodes.
>
> Also, some studies focus on object segmentation that can extract objects and symbols from images [6]. The segmentation method has been applied to research of symbolic model for RL [7]. To conclude, it is possible to extend our framework to various complex domains.

---

> > ### Comment · Reviewer_n4PR · 2023-11-22
> >
> > Thanks for your response, which I'm largely happy with.
> >
> > > Our framework focuses on deterministic, fully observable settings in the integer domain...
> >
> > The "integer domain" part ought to be made explicit in the paper. (Correct me if I'm wrong, but I can't see it anywhere.) This is quite a big assumption, and while I understand that the proposed approach is just the first step in a novel direction, this limitation really ought to be acknowledged in an upfront manner.

---

> > > ### Author Response · Authors · 2023-11-22
> > > **Re: Official Comment by Reviewer n4PR**
> > >
> > > We thank the reviewer for recognizing our rebuttal and for the timely response. We fully agree with the reviewer that the assumption of the integer domain should be acknowledged in the paper. We have revised the paper to explicitly mention this assumption in Section 7. The revised paper also discusses the potential extensions to incorporate more complex domains. The revised paragraph is as follows.
> > >
> > > > This work extensively evaluates the proposed framework on deterministic, fully observable environments within the integer domain. To extend our proposed framework to complex domains, such as continuous input, we can potentially leverage recent studies that have developed genetic programming for different variable types (Virgolin et al., 2017; 2021).

---

> > > > ### Author Response · Authors · 2023-11-23
> > > > **Looking forward to your feedback and discussion**
> > > >
> > > > Dear Reviewer n4PR,
> > > >
> > > > We sincerely thank you for putting so much effort into helping us improve our submission. We hope our rebuttal and the revised paper, which cover the discussions on learning action planning models, our assumptions, tasks, demonstrations, critical actions, etc., have addressed your concerns. As the deadline for the rebuttal period is approaching, we very much look forward to your feedback. We are more than happy to discuss any further questions with you.
> > > >
> > > > Thanks,
> > > >
> > > > Authors

---

> > > > ### Comment · Reviewer_n4PR · 2023-11-23
> > > >
> > > > Thanks for this. While I do find the integer domain assumption to be quite restrictive, you've proposed an interesting, novel approach and it's promising that there are potential avenues to reducing the assumptions going forward. I've updated my score accordingly.

---

> > > > > ### Author Response · Authors · 2023-11-23
> > > > > **Re: Official Comment by Reviewer n4PR**
> > > > >
> > > > > We sincerely thank the reviewer for the timely response and increasing the score.

---

> ### Author Response · Authors · 2023-11-21
> **Response to Reviewer n4PR (2/2)**
>
> > The assumptions around "critical actions" aren't clear. It seems that the approach requires a human expert to deem which state variables are important (e.g., those corresponding to the inventory in Minecraft) and which are non-critical (e.g., the player's position). Again, this ought to be discussed.
>
> We agree with the reviewer that our framework requires a human expert to decide the choice of state variables. The coverage of critical actions depends on the selection of the effect space. We note that the effect space is not unique. In this paper, we consider the player’s position non-critical, which is arguable. The player’s position could be critical if we include the variables of position. In real-world scenarios, many works assume that the sensor input will be transferred to state variables [7], which also requires a human expert to determine the variables.
>
> > A secondary concern is that there doesn't appear to be any source code included with the submission (correct me if I'm wrong) and the reinforcement learning setup isn't clear.
>
> We thank the reviewer for raising the concern. We will provide the file of the source code in the material. The implementation details that the reviewer asked for are listed as follows.
> - The grid world representation is a 3x8x8 image with numeric values, where three channels represent different properties (object type, color, status).
> - We apologize for the misleading information. The reward was paid only at episode termination. Thanks for pointing out the problem.
> - A convolutional neural net and a fully connected network are trained. The kernel size of the first layer in CNN is 3x3, and the others are 2x2.
> - Let the sufficient demonstrations be in the original domain and few-shot demonstrations in the variant domain. All methods are re-trained in the variant domain before pretraining. The pretraining of each method is set as follows:
> Our method uses the demonstrations in the original domain for induction and the demonstrations in the variant domain for re-induction.
>   - BC-PPO uses all demonstrations for behavior cloning.
>   - GAIL uses all demonstrations for the discriminator in training.
>
> The implementation details are elaborated in the appendix, and we will revise the appendix if the information is not clear.
>
>
> > I'm familiar with much of the related work on hierarchical task learning, and while **I'm not 100% sure about the authors' claim that "induc(ing) the required subtasks has not been addressed" (for example, Icarte et al., 2019 arguably do this to some degree)**, their approach to the problem is novel. I certainly haven't seen genetic programming applied in this context before, and it's an interesting idea!
>
> We thank the reviewer for the appreciation, and we apologize for the overreach statement. While most of the works assume pre-defined subtasks, some studies have developed approaches to learning the subtasks from data to some degree. We will revise the paper and correct our statement. Following is the revision:
>
>
> … Despite the success in building hierarchical models shown in previous works, these works put less emphasis on inducing subtask rules and substructure. Therefore, we develop a method to induce symbolic knowledge and leverage it for hierarchical task representation.
>
> ### References
>
> [1] William et al. “Genetic Programming with Epigenetic Local Search” Genetic and Evolutionary Computation Conference 2015
>
> [2] Marco et al.  “Improving Model-Based Genetic Programming for Symbolic Regression of Small Expressions” Evolutionary Computation 2021
>
> [3] Evans et al. “Evolutionary Deep Learning: A Genetic Programming Approach to Image Classification” Congress on Evolutionary Computation 2018
>
> [4] Virgolin et al. “Improving Model-based Genetic Programming for Symbolic Regression of Small Expressions” Evolutionary Computation 2021
>
> [5] Virgolin et al. "Scalable genetic programming by gene-pool optimal mixing and input-space entropy-based building-block learning" Genetic and Evolutionary Computation Conference 2017
>
> [6] Chen et al. "Unsupervised object segmentation by redrawing" NeurIPS 2019
>
> [7] Hasanbeig et al. “DeepSynth: Automata Synthesis for Automatic Task Segmentation in Deep Reinforcement Learning” AAAI 2021
>
> ### Conclusion
>
> We are incredibly grateful to the reviewer for the detailed and constructive review. We believe our responses address the concerns raised by the reviewer. Please kindly let us know if there are any further concerns or missing experimental results that potentially prevent you from accepting this submission. We would be more than happy to address them if time allows. Thank you very much for all your detailed feedback and the time you put into helping us to improve our submission.

---

### Official Review · Reviewer_9Jkh · 2023-10-31

**Soundness:** 3 good
**Presentation:** 2 fair
**Contribution:** 3 good
**Rating:** 6
**Confidence:** 4

**Summary:**

This paper introduces a framework that combines deep reinforcement learning with classical planning techniques to tackle sparse-reward, goal-directed tasks. The framework utilizes genetic programming to automatically induce task structures from expert demonstrations, enabling efficient and robust learning in various domains. Experimental results demonstrate the framework's superior performance in terms of sample efficiency, task performance, and generalization. The paper also discusses the potential for future enhancements, particularly in addressing stochastic or partially observable environments.

**Strengths:**

The paper presents an innovative approach that integrates deep reinforcement learning with classical planning techniques, offering a unique solution to address sparse-reward, goal-directed tasks. This integration leverages the strengths of both paradigms. The paper provides empirical evidence through experiments in gridworld environments, demonstrating the superior performance of the proposed framework in terms of sample efficiency and task performance compared to other methods, including DRL and imitation learning. Finally, the paper offers a clear and comprehensive discussion of the proposed framework, including its components and the potential for future enhancements, such as handling probabilistic rules in uncertain environments.

**Weaknesses:**

- There is a plethora of works on integrating learning and planning, [1,2,3] to name a few. Missing them in the related work section devalues the overall presentation of the work. I strongly suggest having this line of work included and compared against.
- In section 3, the MDP tuple is not explained. By convention, one could understand what `S` , `A`, and others stand for, but it would be better if authors could include their formal definitions.
- How critical states and actions are recognized? I see that it is discussed in section 4, but there are no verifications for such assumptions in the paper.
- Most recent method used as a baseline was introduced in 2020, which would neglect the progress made in the literature since then. I would suggest to include more recent approaches as baselines as well.
- With the use of genetic algorithms, since it is one the most important integral parts of the proposed method, how would it affect the scalability of the overall approach?


[1] Veloso, Manuela, et al. "Integrating planning and learning: The PRODIGY architecture." Journal of Experimental & Theoretical Artificial Intelligence 7.1 (1995): 81-120.
[2] Danesh, Mohamad Hosein, Panpan Cai, and David Hsu. "LEADER: Learning Attention over Driving Behaviors for Planning under Uncertainty." *Conference on Robot Learning*. PMLR, 2023.
[3] Zhao, Luyang, et al. "PLRC\*: A piecewise linear regression complex for approximating optimal robot motion." *2020 IEEE/RSJ International Conference on Intelligent Robots and Systems (IROS)*. IEEE, 2020.

**Questions:**

See above.

---

> ### Author Response · Authors · 2023-11-21
> **Response to Reviewer 9Jkh (1/2)**
>
> We sincerely thank the reviewer for the thorough and constructive comments. Please find the response to your questions below.
>
> ### Responses to Questions
>
> > How does the proposed approach favor in comparison with option discovery approaches that directly learn useful substructures? What are your thoughts on the comparison of their sample efficiency?
>
> Thanks for the suggestion. The mentioned works broadly involve the concept of the learning-planning framework. In these studies, “planning” is an umbrella term for a wide range of search methods, such as A* or tree search. Our work focuses on classical planning with action models; we believe that our paper sufficiently discusses relevant works in this field. We listed the summary of the mentioned papers below.
>
> [1] introduced an architecture that combines planning with learning modules to reduce planning time, improve solution quality, and refine domain knowledge. In this method, the control rules are pre-defined by human experts, while our method induces the rules from demonstrations.
>
> [2] leverages the POMDP planner, which applies belief tree search with heuristics and learns importance distribution for the sample-based planner.
>
> [3] uses combining A* search with regression to create a hybrid search space representation for robot motion, which is relatively irrelevant to our work.
>
> We have revised Section 2 to mention [1, 2] in the related works, marked in blue in the manuscript, and have thoroughly discussed and acknowledged the related work on learning planning action models as follows.
>
> **Learning planning action models.** Some works have focused on integrating learning and planning to enhance capabilities in complex environments (Danesh et al., 2023; Veloso et al., 1995). To leverage the strategies of classical planning, many works have developed building planning action models, including skill acquisition and action schema learning (Arora et al., 2018; Stern & Juba, 2017; Pasula et al., 2007; Callanan et al., 2022; Silver et al., 2022; Mao et al., 2022; Yang et al., 2007). However, these works mainly focus on the existing planning benchmark and less focus on general Markov decision process (MDP) problems. On the other hand, to address the issue of sample efficiency in DRL, several techniques explored the integration of symbolic knowledge with DRL by learning planning action models (Jin et al., 2022; Lyu et al., 2019). In this work, we aim to bridge this gap by extending these approaches to incorporate inferred knowledge into DRL, enhancing its applicability in complex decision-making scenarios.
>
> > In section 3, the MDP tuple is not explained. By convention, one could understand what S, A, and others stand for, but it would be better if authors could include their formal definitions.
>
> We thank the reviewer for the suggestion. We previously formally introduced MDP tuples and related notations in the appendix. As suggested by the reviewer, we will revise the paper and move it to the main paper.
>
> >  How critical states and actions are recognized? I see that it is discussed in section 4, but there are no verifications for such assumptions in the paper.
>
> In our work, we follow Lee et al. [4] to define critical action. In Lee et al., proper abstraction mapping is defined as a low-level action is whether changes the state at the abstraction level or remains the same, and we adopt the mapping and consider the action in this mapping. We assume that proper abstraction exists in the MDP problems that there is a mapping to a planning domain, and action models can be induced by the proposed framework. We also revised the paper and made the definition clear.
>
> > Most recent method used as a baseline was introduced in 2020, which would neglect the progress made in the literature since then. I would suggest to include more recent approaches as baselines as well.
>
> We compared our proposed method to various approaches that are widely considered state-of-the-art in deep RL (PPO), imitation learning (GAIL), and RL exploration (RIDE). We also included comparisons to these state-of-the-art approaches incorporated with advanced techniques, e.g., BC-PPO, and DQN with replay buffer spiking. We believe the comparisons are sufficient to verify the effectiveness of the proposed method. We would appreciate it if the reviewer could be more specific about which recent methods the reviewers want us to compare against.

---

> ### Author Response · Authors · 2023-11-21
> **Response to Reviewer 9Jkh (2/2)**
>
> > With the use of genetic algorithms, since it is one the most important integral parts of the proposed method, how would it affect the scalability of the overall approach?
>
> We thank the reviewer for raising this question. We adopt genetic programming (GP) to address non-binary values in states since GP has a more flexible model compared to other regression models such as decision trees or linear regression. Also, it has more expressibility than numerical regression methods. Besides the integer domain we use in our work, previous studies also demonstrate the efficacy of GP on continuous or binary domains. For instance, [5] focuses on symbolic regression using genetic programming on real-world benchmark datasets with continuous values, while [6] works on Boolean circuit benchmark problems. Therefore, it can extend to various domains once the variable types are known.
>
> ### References
>
> [1] Veloso et al. "Integrating planning and learning: The PRODIGY architecture" Journal of Experimental & Theoretical Artificial Intelligence 7.1 (1995): 81-120.
>
> [2] Danesh et al. "LEADER: Learning Attention over Driving Behaviors for Planning under Uncertainty" Conference on Robot Learning 2023
>
> [3] Zhao, Luyang, et al. "PLRC*: A piecewise linear regression complex for approximating optimal robot motion" IEEE/RSJ International Conference on Intelligent Robots and Systems 2020
>
> [4] Lee et al. “AI planning annotation in reinforcement learning: Options and beyond” In Planning and Reinforcement Learning Workshop at ICAPS 2021
>
> [5] Virgolin et al. “Improving Model-based Genetic Programming for Symbolic Regression of Small Expressions” Evolutionary Computation 2021
>
> [6] Virgolin et al. "Scalable genetic programming by gene-pool optimal mixing and input-space entropy-based building-block learning" Genetic and Evolutionary Computation Conference 2017
>
> ### Conclusion
>
> We are incredibly grateful to the reviewer for the detailed and constructive review. We believe our responses address the concerns raised by the reviewer. Please kindly let us know if there are any further concerns or missing experimental results that potentially prevent you from accepting this submission. We would be more than happy to address them if time allows. Thank you very much for all your detailed feedback and the time you put into helping us to improve our submission.

---

> > ### Comment · Reviewer_9Jkh · 2023-11-22
> > **Response to authors**
> >
> > I would like to thank the authors for their thorough response. They have addressed my concerns and questions about the paper. After carefully going through the discussions between authors and other reviewers, I intend to increase my rating from 5 to 6.

---

> > > ### Author Response · Authors · 2023-11-22
> > > **Re: Response to authors**
> > >
> > > We sincerely thank the reviewer for the timely response and increasing the score.

---

### Official Review · Reviewer_Yjtd · 2023-11-01

**Soundness:** 3 good
**Presentation:** 3 good
**Contribution:** 3 good
**Rating:** 8
**Confidence:** 4

**Summary:**

- The proposed work automatically infers substructures from demonstrations that enable generalization to adapt to new environments.

- Induction stage: The approach uses genetic programming to infer action schemata and extracts symbolic knowledge in the form of critical actions from it.

- Training stage: The approach uses the inferred model of symbolic rules that represent task substructures to build a critical action network online from the goal by backward-chaining. Whenever critical actions are achieved instrinsic rewards are provided.

- The work considers deterministic, fully observable but unknown environments.

**Strengths:**

- The idea of using genetic programming to learn symbolic action schema and extracting the relevant substructures for a specific goal is interesting.

- The paper is well-written and clear. The results are convincing.

**Weaknesses:**

- There is much work on learning action models from demonstrations or trajectories as also mentioned in the related work section. How is the induction stage different from that line of research?

- In the experimental section, it will help to report the critical actions inferred by the approach for different domains.

- How does the proposed approach favor in comparison with option discovery approaches that directly learn useful substructures? What are your thoughts on the comparison of their sample efficiency?

**Questions:**

Included with the limitations.

---

> ### Author Response · Authors · 2023-11-21
> **Response to Reviewer Yjtd**
>
> We sincerely thank the reviewer for the thorough and constructive comments. Please find the response to your questions below.
>
> ### Responses to Questions
>
> > There is much work on learning action models from demonstrations or trajectories as also mentioned in the related work section. How is the induction stage different from that line of research?
>
> Most of the works on learning action models from demonstrations consider the planning benchmark, while our work extends the concepts to the RL domain. The representation of the RL benchmark is different from those in classical planning. We adopt several techniques to map the representation, such as the formulation of action schemata and the use of genetic programming for non-binary values. These improvements enable our framework to work on MDP directly, which considers the variable types from MDP and the lower level of abstraction closer to the DRL domain.
>
> > In the experimental section, it will help to report the critical actions inferred by the approach for different domains.
>
> Thanks for the suggestion. We will complete the figures and the table of the critical actions and its graph for all domains in the appendix. Here we display the critical actions of Doorkey
>
> | Action name | MDP action | Effect | Precondition |
> | --- | --- | --- | --- |
> | pickup_key | pickup | has_key = 1, at_key=0 | at_key = 1 |
> | open_door | toggle | door_state=2 | at_door=1, door_state=0 |
> | turn_on | toggle | next_switch=2 | door_state=2, at_switch=1 |
>
>
> > How does the proposed approach favor in comparison with option discovery approaches that directly learn useful substructures? What are your thoughts on the comparison of their sample efficiency?
>
> Option discovery approaches assume that there exists a policy for each subtask and an option that can assign the policy according to the current state. It requires pre-trained policies and pre-defined subtasks. In contrast, our method induces action models from demonstrations or trajectories with the assumption of variables. The configuration is similar to the program synthesis approach which assumes domain-specific language.
>
> Generally, the sub-policies in hierarchical reinforcement learning come from two methods — using pre-defined subtasks or learning subtasks from exploration.
>
> Training the option with pre-defined policies performs higher sample efficiency since it has stronger assumptions about the context of the subtask, while our method involves induction of the subtask substructure. In addition, it is inefficient if the pre-defined subtask is incorrect.
>
> On the other hand, some methods build the sub-policies and the option from exploration, which does not require pre-defined sub-policies. While they train a set of induced sub-policies to accomplish different subtasks, our framework aims to guide one RL agent to learn a whole sequential task. These two settings are specified for different scenarios. Although learning sub-policies may be more efficient due to the complexity, it requires additional resources to train the meta-controller. In addition, HRL can not ensure the overall policy is optimal even if the subpolicies are optimal. In contrast, our method uses intrinsic reward to guide a single agent, which is more flexible to learn the policies.
>
>
> ### Conclusion
>
> We are incredibly grateful to the reviewer for the detailed and constructive review. We believe our responses address the concerns raised by the reviewer. Please kindly let us know if there are any further concerns or missing experimental results. We would be more than happy to address them if time allows. Thank you very much for all your detailed feedback and the time you put into helping us to improve our submission.

---

> > ### Comment · Reviewer_hQ37 · 2023-11-22
> >
> > Dear Authors,
> > I need more time to revisit the revised version of the paper and the details in the response.
> > Thanks very much for your response and the updates to the paper.

---

### Official Review · Reviewer_hQ37 · 2023-11-02

**Soundness:** 2 fair
**Presentation:** 3 good
**Contribution:** 2 fair
**Rating:** 5
**Confidence:** 4

**Summary:**

This paper presents a method that integrates automated planning and deep reinforcement learning for solving sequential decision-making problems, such as grid navigation problems.
This paper assumes that the state of an RL environment is annotated with symbolic representations, such that the agent can obtain a trace of symbolic states given an RL trajectory.
The main idea is that the agent first learns an action model, which is called critical action in this paper, that specifies how the state changes due to the application of the action, given human demonstrations. After obtaining an action model in which a symbolic action is associated with a single RL action, the agent performs planning with backward chaining to find a plan and use it to train an RL policy with PPO.

**Strengths:**

* Originality: The originality of this work is the notion of action-effect linkage that finds effect variables for an action. For finding symbolic expression of action models, this paper proposes a method using genetic programming to find effects with numeric expressions.
* Quality: The paper shows ideas in words and figures to help to understand the basic concepts.
* Significance: This paper shows that the proposed method outperforms several baselines in grid-world environments.

**Weaknesses:**

Originality:

Regarding action model acquisition from RL environments, there are related works, such as [1, 2]. Learning action models from symbolic traces is also a widely studied topic in automated planning [3, 4]. The papers such as [1, 2] have similar problem settings, so it is worth mentioning in the paper.

[1] Jin, M., Ma, Z., Jin, K., Zhuo, H. H., Chen, C., & Yu, C. (2022, June). Creativity of ai: Automatic symbolic option discovery for facilitating deep reinforcement learning. In Proceedings of the AAAI Conference on Artificial Intelligence (Vol. 36, No. 6, pp. 7042-7050).

[2] Lyu, D., Yang, F., Liu, B., & Gustafson, S. (2019). SDRL: Interpretable and Data-Efficient Deep Reinforcement Learning Leveraging Symbolic Planning.

[3] Arora, A., Fiorino, H., Pellier, D., Métivier, M., & Pesty, S. (2018). A review of learning planning action models. The Knowledge Engineering Review, 33, e20.

[4] Yang, Q., Wu, K., & Jiang, Y. (2007). Learning action models from plan examples using weighted MAX-SAT. Artificial Intelligence, 171(2-3), 107-143.

Clarity:
* There are very strong assumptions made in the paper that can be easily overlooked.
For example, action precondition is defined over all variables, one action impacts specific features, and critical action maps to exactly one MDP action. It needs clarification on why those assumptions are needed.

Significance:

Environments tested in the experiment section are small-scale.

**Questions:**

### General questions

1. In the experiments, can you show the number of action operators learned to solve each problem? What is the length of the plans? For generating demonstrations given a problem environment, what are the prior knowledge required for an expert?

2. Do you assume that you have a symbolic state annotation for an RL state and grounded action operators in the environment?


### Learning action models from traces
3. In the action-effect linkages, is $a$ an RL action (for example, make1 in Figure 3(b))?  How do we know that make1 maps to make_stick ? For computing mutual information, how many cases need to be considered? Is it |A| * \sum_{r=1^n} nCr for n being the number of variables? nCr is the number of selecting r variables from n variables. What is the typical number of variables considered in the experiment?

4. Precondition is found by learning a decision tree. I think it will be meaningful to compare this approach of learning action model (effect and precondition) with papers mentioned above [1,2] or other methods in automated planning [3, 4].

5. In the Minecraft example, make_stick fits well with the presented approach. make_stick only changes features to make a stick by consuming wood and stick in the inventory, and there is a unique MDP action make1 that corresponds to this action. Minecraft environment will generate a sequence of actions that transform objects to a particular object and it will be given by human demonstration. What aspect of this example can be generalized such that the agent doesn't need to see the demonstration?

6. What are the critical actions in mini-grid environments?

### Planning
7. What if some of the critical actions had incorrect preconditions and effects learned? Figure 6 shows that the accuracy cannot reach 100% and it implies it may miss some effect variable. Or it may skip some states that some critical action is applicable due to incorrect precondition. Is there any guarantee that the presented approach solves the problem?

8. What is the length of the plans for solving each problem?

### Reinforcement Learning

9. DoorKey or 4-Rooms environments are too small-scale. Did you try on larger scale problems in mini-grid or BabyAI?

10. Did you train a single RL agent for low-level policy?

---

> ### Author Response · Authors · 2023-11-21
> **Response to Reviewer hQ37 (1/4)**
>
> We sincerely thank the reviewer for the thorough and constructive comments. Please find the response to your questions below.
>
> ### Responses to Questions
>
> > Regarding action model acquisition from RL environments, there are related works, such as [1, 2]. Learning action models from symbolic traces is also a widely studied topic in automated planning [3, 4]. The papers such as [1, 2] have similar problem settings, so it is worth mentioning in the paper.
>
> We thank the reviewer for suggesting these relevant works on learning planning action models. We did mention [3] in Section 2 to discuss the learning planning action model. We include the rest of the mentioned works and revise the main paper to discuss them. The revised paragraph is as follows:
>
> **Learning planning action models.** Some works have focused on integrating learning and planning to enhance capabilities in complex environments (Danesh et al., 2023; Veloso et al., 1995). To leverage the strategies of classical planning, many works have developed building planning action models, including skill acquisition and action schema learning (Arora et al., 2018; Stern & Juba, 2017; Pasula et al., 2007; Callanan et al., 2022; Silver et al., 2022; Mao et al., 2022; Yang et al., 2007). However, these works mainly focus on the existing planning benchmark and less focus on general Markov decision process (MDP) problems. On the other hand, to address the issue of sample efficiency in DRL, several techniques explored the integration of symbolic knowledge with DRL by learning planning action models (Jin et al., 2022; Lyu et al., 2019). In this work, we aim to bridge this gap by extending these approaches to incorporate inferred knowledge into DRL, enhancing its applicability in complex decision-making scenarios.
>
> > There are very strong assumptions made in the paper that can be easily overlooked. For example, action precondition is defined over all variables, one action impacts specific features, and critical action maps to exactly one MDP action. It needs clarification on why those assumptions are needed.
>
> We thank the reviewer for raising this concern, which helps us clarify our work. The assumptions and why they are needed are discussed below. We will revise the paper to include the discussions.
>
> - **Action precondition is defined over all variables:** In this work, we define the effect space and precondition space, and the effect space decides the granularity of the critical actions. While we also take the precondition space into account, it is a redundant assumption and is misleading. It is clearer if assumptions about preconditions are removed. We thank the reviewer for pointing out the problem and apologize for the confusion. We have revised the formulation and removed the assumption of precondition space.
> - **One action impacts specific features**: We assume that one action impacts specific features and consider the case if the case does not hold. First, if more than one action impacts the same features, due to our assumption of critical action maps to exactly one MDP action, it will be considered as two critical actions. Second, if one action may impact different features due to the condition, it will also be considered as two critical actions, and different preconditions are induced.
> - **Critical action maps to exactly one MDP action:** We refer to the proper abstraction in the study of Lee et al. that the abstraction is proper if an action remains in the same abstraction state or changes to another state. The concept is similar to “representative frames” in computer vision. However, in some scenarios, a meta-action may map to multiple sequential MDP actions. We also have revised Section 3 to make our definition clearer.
>
> > Environments tested in the experiment section are small-scale.
> > DoorKey or 4-Rooms environments are too small-scale. Did you try on larger scale problems in mini-grid or BabyAI?
>
> We would like to emphasize that our experiments adopt tasks that are considered challenging in recent works [5-8]. Specifically, the Switch tasks require an agent to turn on the switches according to a specific order, which requires acquiring long-horizon behaviors. The difficulties of the Switch tasks are demonstrated in the experimental results. For instance, all the RL and imitation learning baselines fail to solve the 8-Switches task. On the other hand, completing the Minecraft task involves making tools with different materials, similar to the setting in [5, 6].  The Doorkey task is a standard benchmark in Mini-grid and has been used in [7, 8].
>
> We understand that "larger scale" may refer to many aspects, such as state space (e.g., grid size), action space, problem difficulties, horizons, etc. We would appreciate it if the reviewer could be more specific about “larger scale problems”.

---

> ### Author Response · Authors · 2023-11-21
> **Response to Reviewer hQ37 (2/4)**
>
> > In the experiments, can you show the number of action operators learned to solve each problem? What is the length of the plans?
>
> We list the details of each environment below.
> - In Doorkey, the number of action operators is 7, the length of the abstract plan is 3 (pickup key $\rightarrow$ open the door $\rightarrow$ turn on the switch), while the number of low-level actions (i.e., the horizons in the MDP) ranges from 10 to 20 per episode.
> - In n-Switch, the number of action operators is 5, the length of the abstract plan is $n$ (turn on switch * n), while the number of low-level actions (i.e., the horizons in the MDP) ranges from 10 to 30 per episode.
> - In Minecraft, the number of action operators is 13. The length of the abstract plan takes 10 to 20 steps, depending on the task. The number of low-level actions (i.e., the horizons in the MDP) ranges from 30 to 100 per episode.
>
> We will revise the paper to include the information.
>
> > For generating demonstrations given a problem environment, what are the prior knowledge required for an expert?
>
> We would like to point out that the demonstrations do not need to be optimal in efficiency, and therefore, we do not need to collect them from an expert. However, we assume that the demonstrations trigger a sequence of subgoals (e.g., pickup key $\rightarrow$ go to the door $\rightarrow$ open the door in Doorkey) based on task-specific prior knowledge. We will discuss this in the revised paper.
>
> > Do you assume that you have a symbolic state annotation for an RL state and grounded action operators in the environment?
>
> - **Symbolic state annotation for an RL state**:
> We assume a set of symbolic state variables for an RL state. For instance, in the Doorkey environment, the state variables include the door status and the key status. We assume that there are mapping functions that transfer raw sensor input to symbolic state variables similar to [1].
>
> - **Grounded action operators**:
> We are unsure whether the reviewer means that we assume grounded action operators must exist in the environment. We do not assume there are grounded action operators directly. We assume the effect space and the critical actions impact the variables in the effect space. However, if the effect space is not set properly, it is possible that no critical action will be induced in the procedure.
>
> > In the action-effect linkages, is $a$ an RL action (for example, make1 in Figure 3(b))? How do we know that make1 maps to make_stick? For computing mutual information, how many cases need to be considered? Is it |A| * \sum_{r=1^n} nCr for n being the number of variables? nCr is the number of selecting r variables from n variables. What is the typical number of variables considered in the experiment?
>
> - **RL action, make1 maps to make_stick**:
> In the action-effect linkages, $a$ is an RL and MDP action, which the agent directly executes. We note that the name is only for the convenience of reference, the framework does not acquire the exact name of the make_stick. It induces the effect variables that are related to making sticks and maps the effect to the RL action make1. We named it make_stick for the explanation in the paper.
> Ref p.5 “Note that we name the critical action ψ for the convenience of reference, which is not known when inducing action schemata.”
> - **Mutual information and typical number of variables**:
> In the experiment, Minecraft tasks include 22 variables, while Doorkey and Switch include less than 10 variables. For computing mutual information, we consider the number of co-occurred effect variables, and we count the effects that have occurred in the demonstrations (N_E), which is much less than nCr. For instance, in Minecraft, the number of cases is |A| x N_E, where |A| is 9 and N_E is 41. The implicit assumption is that the demonstrations should have at least cover one sample.

---

> ### Author Response · Authors · 2023-11-21
> **Response to Reviewer hQ37 (3/4)**
>
> > In the Minecraft example, make_stick fits well with the presented approach. make_stick only changes features to make a stick by consuming wood and stick in the inventory, and there is a unique MDP action make1 that corresponds to this action. Minecraft environment will generate a sequence of actions that transform objects to a particular object and it will be given by human demonstration. What aspect of this example can be generalized such that the agent doesn't need to see the demonstration?
>
>  We consider the possibility of differences between RL actions; therefore, we distinguish the critical actions with the same effects but with different RL actions. If a ground truth critical action can be mapped to two actions, the module is supposed to induce two critical actions with the same effects and preconditions, and both can be induced by backward chaining.
>
> In addition, our framework allows for learning with trajectories collected during training instead of demonstrations. To test the efficacy of our framework without demonstrations, we train a PPO agent from scratch and collect data online in Minecraft MULTIPLE task. During training, we (re-)induce the action model every 40k steps, sampling 80k transition data for induction. We found that the main difference is the accuracy of genetic programming, which is listed as follows:
> | Steps | Accuracy |
> | --- | --- |
> | 40k | 47 $\pm$ 6% |
> | 80k | 49 $\pm$ 5% |
> | 120k | 49 $\pm$ 6% |
> | 160k | 48 $\pm$ 6% |
> | 200k | 49 $\pm$ 5% |
>
> The setting of accuracy is the same as the ablation study, where we calculate the correct rules out of 27 effect rules in Minecraft. The result shows that we can induce 50% of rules using the data collected by the agent learning from scratch. To be more specific, the rules in basic subtasks such as making a stick and picking up a wood are correct, while for advanced subtasks such as making an enhance table, GP overfits due to insufficient samples. This can be improved by adopting exploration-based methods, such as ICM or RND which are left for our future work.
>
> > What are the critical actions in mini-grid environments?
>
> What critical actions will be induced depends on the effect space and the state variables. For example, primitive moving actions (e.g., Up, Down, Left, Right) are not considered as critical actions according to our choice of effect space and precondition space. In contrast, if we include the position of the agent in the effect space, primitive moving actions will be considered as critical actions.
>
> > What if some of the critical actions had incorrect preconditions and effects learned? Figure 6 shows that the accuracy cannot reach 100% and it implies it may miss some effect variable. Or it may skip some states that some critical action is applicable due to incorrect precondition. Is there any guarantee that the presented approach solves the problem?
>
> As mentioned by the reviewer, genetic programming can fail to find the global optimal solution. Therefore, it may skip some states and lead to an incomplete network if the precondition and the effect are incorrect. In the worst case when all the critical actions are missed, as discussed in Section 7, the performance of our proposed framework would be equivalent to deep RL methods used to learn the low-level policy.
>
> > Did you train a single RL agent for low-level policy?
>
> In our experiments, we trained PPO and DQN policies as baseline low-level policies. Our proposed framework also involves learning a low-level policy. To generate demonstrations, we used heuristic programs instead of learning policies for each task.

---

> ### Author Response · Authors · 2023-11-21
> **Response to Reviewer hQ37 (4/4)**
>
> ### References
>
> [1] Jin et al. “Creativity of ai: Automatic symbolic option discovery for facilitating deep reinforcement learning” AAAI 2022
>
> [2] Lyu et al. “SDRL: Interpretable and Data-Efficient Deep Reinforcement Learning Leveraging Symbolic Planning” AAAI 2019
>
> [3] Arora et al. "A review of learning planning action models" The Knowledge Engineering Review 2018
>
> [4] Yang et al. “Learning action models from plan examples using weighted MAX-SAT” Artificial Intelligence 2007
>
> [5] Furelos-Blanco et al. "Hierarchies of Reward Machines" ICML 2023
>
> [6] Sohn et al. "Hierarchical reinforcement learning for zero-shot generalization with subtask dependencies" NeurIPS 2018
>
> [7] Wan et al. “DEIR: Efficient and Robust Exploration through Discriminative-Model-Based Episodic Intrinsic Rewards” IJCAI 2023
>
> [8] Liu et al. “Hierarchical Programmatic Reinforcement Learning via Learning to Compose Programs” ICML 2023
>
> ### Conclusion
>
> We are incredibly grateful to the reviewer for the detailed and constructive review. We believe our responses address the concerns raised by the reviewer. Please kindly let us know if there are any further concerns or missing experimental results that potentially prevent you from accepting this submission. We would be more than happy to address them if time allows. Thank you very much for all your detailed feedback and the time you put into helping us to improve our submission.

---

> > ### Author Response · Authors · 2023-11-23
> > **Looking forward to your feedback and discussion**
> >
> > Dear Reviewer hQ37,
> >
> > We sincerely thank you for putting so much effort into helping us improve our submission. We hope our rebuttal and the revised paper, which cover the discussions on learning action planning models, our assumptions, tasks, demonstrations, critical actions, etc., have addressed your concerns. As the deadline for the rebuttal period is approaching, we very much look forward to your feedback. We are more than happy to discuss any further questions with you.
> >
> > Thanks,
> >
> > Authors

---

### Author Response · Authors · 2023-11-22
**Paper Revision**

We would like to sincerely thank all the reviewers for their thorough and constructive comments. We have revised our paper to incorporate them marked as blue. The major changes are summarized as follows:

(1) **[Related work]** We thank the reviewers for pointing out many relevant works and we have revised the related work section to include the papers in the following directions:

- *action model acquisition from RL environments and from symbolic trace* (Reviewer hQ37)
- *integrating learning and planning* (Reviewer 9Jkh)

(Reviewer n4PR) We also revised the overreach statement about the issue of inducing subtasks in previous works.

(2) **[Critical action definition and clarification]** (Reviewer 9Jkh, Reviewer hQ37) We refer to the proper abstraction in Lee et al. to define critical actions. In the original paper, the description is at the end of Section 4. We reorganized the section and put this part to the front to clarify the definition of critical actions.


(3) **[Removal of the precondition space]** (Reviewer hQ37) The assumption of the precondition space is redundant, and we have removed the statement.

(4) **[Assumption of the integer domain]** (Reviewer n4PR) We have explicitly indicated the assumption of the integer domain in Section 7. In addition, we add the discussion of the potential extension to more complex domains.

(5) **[DRL module]** (Reviewer n4PR) We have completed the implementation detail in Appendix B.4, which presents the neural network architecture and setup of the PPO agent used in our framework and as baselines.

(6) **[Critical action]** (Reviewer Yjtd) We have added the induced critical actions and the networks in the experiments in Appendix D.

(7) **[Reward function]** (Reviewer n4PR) We revised the reward function in Appendix B.5 to make it clear for reading in Appendix B.5.

(8) **[Others]** We have fixed the grammatical errors and made some statements more concise for the page limit in the revision.

---

### Meta-Review · Area_Chair_NoX2 · 2023-12-06

**Metareview:**

This paper introduces a framework that combines deep reinforcement learning with classical planning techniques to tackle sparse-reward, goal-directed tasks. The framework infers PDDL-style preconditions and effects of "critical actions" (actions that are crucial for success and must be executed in a particular order) via genetic programming from expert demonstrations. Experimental results in three gridworld environments demonstrate that the proposed method has better task performance and provides the ability to transfer learnt behavior to new task instances. The authors have incorporated many of the reviewers suggestions, to improve clarity, but some questions about scalability of the approach remain.

**Justification For Why Not Higher Score:**

Scalability of proposed mechanism has not been sufficiently addressed. Somewhat hard to understand contributions in comparison to prior work.

**Justification For Why Not Lower Score:**

Presents a novel algorithm that integrates automated planning and deep reinforcement learning, provides empirical results to show that the algorithm can be useful in certain situations.

---

### Decision · Program_Chairs · 2024-01-16

Accept (poster)